# Binary outcomes of enhancer activity underlie stable random monoallelic expression

**Djem U Kissiov[1], Alexander Ethell[1], Sean Chen[1], Natalie K Wolf[1], Chenyu Zhang[1], Susanna M Dang[1], Yeara Jo[1], Katrine N Madsen[1], Ishan Paranjpe[1], Angus Y Lee[2], Bryan Chim[3], Stefan A Muljo[3], David H Raulet[1]\***

[1]Division of Immunology and Pathogenesis, Department of Molecular and Cell Biology, University of California, Berkeley, Berkeley, United States; [2]Cancer Research Laboratory, University of California, Berkeley, Berkeley, United States; [3]Laboratory of Immune System Biology, National Institute of Allergy and Infectious Diseases, National Institutes of Health, Bethesda, United States

**Abstract** Mitotically stable random monoallelic gene expression (RME) is documented for a small percentage of autosomal genes. We developed an in vivo genetic model to study the role of enhancers in RME using high-resolution single-cell analysis of natural killer (NK) cell receptor gene expression and enhancer deletions in the mouse germline. Enhancers of the RME NK receptor genes were accessible and enriched in H3K27ac on silent and active alleles alike in cells sorted according to allelic expression status, suggesting enhancer activation and gene expression status can be decoupled. In genes with multiple enhancers, enhancer deletion reduced gene expression frequency, in one instance converting the universally expressed gene encoding NKG2D into an RME gene, recapitulating all aspects of natural RME including mitotic stability of both the active and silent states. The results support the binary model of enhancer action, and suggest that RME is a consequence of general properties of gene regulation by enhancers rather than an RME-specific epigenetic program. Therefore, many and perhaps all genes may be subject to some degree of RME. Surprisingly, this was borne out by analysis of several genes that define different major hematopoietic lineages, that were previously thought to be universally expressed within those lineages: the genes encoding NKG2D, CD45, CD8α, and Thy-1. We propose that intrinsically probabilistic gene allele regulation is a general property of enhancer-controlled gene expression, with previously documented RME representing an extreme on a broad continuum.

\*For correspondence: raulet@berkeley.edu

Competing interest: The authors declare that no competing interests exist.

## Editor's evaluation

Kissiov et al., show that enhancers can play an instructive role in controlling stable random monoallelic expression (RME). In order to do so, they initially focus on a limited set of natural killer (NK) receptor genes that are subject to RME, which they investigate using several in vivo genetic models. Furthermore, they show that RME can be considerably more prevalent than previously thought, applying to other gene function than receptors and independently of promoter CpG content. Finally, they provide evidence that enhancer strength and/or number might influence the extent of RME, by affecting the probability of promoter activation. Overall, this is a highly relevant manuscript with major implications in gene regulation and enhancer biology and, thus, of broad scientific interest.

## Introduction

In most cases, both alleles of autosomal genes are co-expressed. In recent years random monoallelic expression (RME) has emerged as an important exception that may apply to ~0.5–10% of expressed genes in a given tissue and has been characterized as the autosomal analog of X-inactivation (*Gendrel et al., 2016*). In RME, different cells of a given type express only one allele, both or neither, and this expression pattern is mitotically stable. Notably, RME genes do not share an overarching unifying feature or function (*Deng et al., 2014*; *Eckersley-Maslin and Spector, 2014*; *Gendrel et al., 2016*; *Gimelbrant et al., 2007*; *Reinius et al., 2016*; *Xu et al., 2017*), and the biological role for RME is in most cases not known. RME is distinct from X-inactivation, genomic imprinting, and allelic exclusion of antigen receptor and odorant receptor genes, in that biallelic expression occurs at an appreciable frequency, and expression is largely stochastic rather than imposed by strict feedback regulatory mechanisms (*Gendrel et al., 2016*).

The molecular determinants of RME are poorly understood, in part because of difficulties analyzing single primary cells (*Reinius and Sandberg, 2015*). Chromatin analysis of RME alleles in primary populations has not been possible due to the difficulty of isolating pure cell populations ex vivo with defined RME expression patterns (*Gendrel et al., 2016*; *Xu et al., 2017*). As a result, previous analyses have been limited to clonal cell lines derived from $F_1$ hybrids, where allelic expression is known and clonally stable (*Eckersley-Maslin et al., 2014*; *Gendrel et al., 2014*; *Levin-Klein et al., 2017*; *Xu et al., 2017*). These analyses revealed that enhancers of RME genes are constitutively accessible irrespective of gene or allelic expression status, whereas promoters are accessible only at active alleles (*Levin-Klein et al., 2017*; *Xu et al., 2017*). Therefore, promoter accessibility, rather than enhancer opening and activation, might be the 'gatekeeper' of RME, whereas enhancers, being constitutively open, were proposed to permit rather than impose expression of RME alleles (*Xu et al., 2017*).

Enhancers may play more than a permissive role in RME, however, in light of evidence that enhancers primarily influence the probability of mitotically stable expression, rather than the amount of expression per cell (*Walters et al., 1995*; *Weintraub, 1988*). In fact, deletions of enhancers resulted in mitotically stable gene variegation in both cell lines and normal tissues—notably *Igh* in B cell hybridomas and *Cd8a* in primary thymocytes, among others (*Ellmeier et al., 2002*; *Garefalaki et al., 2002*; *Ronai et al., 1999*; *Sleckman et al., 1997*; *Walters et al., 1995*; *Weintraub, 1988*; *Xu et al., 1996*). Collectively, these data support the binary or 'on/off' model of enhancer action (*Blackwood and Kadonaga, 1998*; *Fiering et al., 2000*), where an increase in enhancer activity at a genetic locus results in an increase in the *probability* of gene expression, rather than an increase in expression per cell. Conversely, weak or reduced enhancer activity results in a lower likelihood of expression, but the cells that express the gene express a similar amount of gene product.

We reasoned that the binary action of enhancers—when limiting—might be a driving principle of RME, and sought to test this in examples of RME with a clear biological purpose: the *Klra* genes (which encode the Ly49 family receptors) and the *Klrc1* gene (which encodes the NKG2A receptor) (*Chess, 2012*; *Eckersley-Maslin and Spector, 2014*; *Gendrel et al., 2016*). These genes, clustered in a ~ 1 Mb stretch of the NK gene complex (NKC) on chromosome 6 in mice, encode cell surface receptors expressed by NK cells that bind MHC I molecules. They are expressed in a variegated (*Raulet et al., 1997*; *Yokoyama et al., 1990*), monoallelic (*Held et al., 1995*), stochastic and largely mitotically stable fashion (*Raulet et al., 2001*), resulting in subpopulations of NK cells that express random combinations of the receptors and consequently exhibit distinct reactivities for cells expressing different MHC I molecules. Regulation of each gene allele is independent, and expression of one *Klra* gene has minimal effects on expression of others (*Tanamachi et al., 2001*). While a clear biological purpose for RME at many genes is lacking, RME at *Klra* genes underlies the basis of the 'missing self' mode of NK cell target recognition (*Kärre et al., 1986*). Furthermore, the system represents a powerful in vivo genetic model of RME, where allelic expression states can be easily assessed at the population level in primary cells using allele-specific antibodies that we and others previously generated (*Tanamachi et al., 2001*; *Vance et al., 2002*), circumventing previous technical limitations to studying RME in single primary cells.

Importantly, competition between *Klra* genes for interaction with a shared enhancer or locus control region is not required for variegation of *Klra* genes, as a *Klra1* (encodes Ly49A) genomic transgene ectopically integrated in different genomic sites was usually expressed with a frequency similar to that of the native *Klra1* gene (~17% of NK cells) (*Tanamachi et al., 2004*). We previously identified a key

DNase I hypersensitive element, $Klra1_{Hss1}$, ~ 5 kb upstream of the *Klra1* gene that is conserved in other *Klra* genes and required for expression of the *Klra1* transgene (*Tanamachi et al., 2004*).

Our central hypothesis is that enhancers, rather than simply being permissive for RME, both limit and directly control the probability of expression of *Klra* genes—and RME alleles generally—in a stochastic and binary fashion. Binary enhancer action, when limiting, may represent a causal mechanism of RME, explaining the pervasiveness of RME across genes and cell types. We have carried out genetic dissection and population analyses to demonstrate that enhancers control the probability of allelic expression and have provided a more general model of the role of enhancers in RME as well as in other developmentally regulated genes.

## Results

### Elements upstream of the *Klra*, *Klrc*, and *Klrk* family genes are transcriptional enhancers with activity in mature NK cells

*Klra* family genes are expressed in a mitotically stable RME fashion by NK cells. Each harbors an accessible chromatin site (Hss1) ~5 kb upstream of the transcription start site (TSS) (*Figure 1A and B*; *Figure 1—figure supplement 1A*). We noticed that related NK receptor genes, including the variegated *Klrc1* gene (encodes NKG2A) and the *Klrk1* gene (encodes NKG2D and is expressed by ~all NK cells), harbor similar elements which we named $Klrc1_{5'E}$ and $Klrk1_{5'E}$, respectively (*Figure 1A, B*; *Figure 1—figure supplement 1A*). All Hss1 and 5′E elements are bound by a similar suite of factors including Runx3, T-bet, and the enhancer-associated acetyltransferase p300 (*Figure 1—figure supplement 1A*).

The $Klra_{Hss1}$ elements were hypothesized to serve as upstream bidirectional promoters active only in immature, developing NK cells (which do not yet express Ly49s); it was proposed that the direction of transcription predetermines the subsequent on or off state of the gene in mature NK cells, which is driven by a distinct downstream promoter (*Saleh et al., 2004*). Recent analysis of *Klra* gene expression in cell lines suggested instead that the $Klra_{Hss1}$ elements are transcriptional enhancers (*Gays et al., 2015*), and that the Hss1 transcripts represent enhancer RNAs (eRNAs). Those conclusions were in turn contested in a subsequent paper (*McCullen et al., 2016*). To address this issue in vivo, we analyzed chromatin features associated with the Hss1 elements in primary NK cells with published ChIP-seq data generated in mature primary splenic NK cells, using the H3K4me1:me3 ratio as an indicator of regulatory element identity (*Calo and Wysocka, 2013*). The Hss1 and 5′E elements are all enriched in H3K4me1 relative to H3K4me3 (*Figure 1A*), indicating enhancer identity. The putative NK receptor gene enhancers all ranked in the top 32% of ATAC-seq accessible peaks with respect to the H3K4me1:me3 ratio. In contrast, known promoters of the respective genes ranked in the bottom 21% (*Figure 1C*). In a deeper analysis, we independently defined enhancers and promoters in mature NK cells. NK cell promoters were defined as previously annotated mouse promoters from the EDPNew database (*Dreos et al., 2017*) enriched in H3K27ac in NK cells, and enhancers were defined as ATAC-seq peaks bound by the p300 histone acetyltransferase that do not overlap with the promoter list. Enhancers defined in this manner were highly skewed to high H3K4me1:me3 ratios, and promoters to low ratios (*Figure 1—figure supplement 1B*). All Hss1 and 5′E elements were classified as enhancers based on the p300-bound enhancer dataset (*Figure 1—figure supplement 1B*). These findings support the conclusion that the Hss1 and 5′E elements elements are enhancers in primary mature NK cells.

To test whether the $Klra7_{Hss1}$ (*Klra7* encodes Ly49G2) and $Klrc1_{5'E}$ enhancers are required in mature NK cells, we adapted a CRISPR/Cas9 nucleofection protocol developed to edit primary human T cells (*Roth et al., 2018*; *Figure 1—figure supplement 2*). We used NK cells from (B6 x BALB/c)$F_1$ hybrid mice and sorted NKG2A[B6]+ or Ly49G2[B6]+ cells using B6-allele reactive monoclonal antibodies against each receptor (*Tanamachi et al., 2001*; *Vance et al., 2002*) in order to follow the fate of a single allele in each case (*Figure 1—figure supplement 2A*). Editing efficiencies of NK cells were lower than that of T cells, resulting in only 30% or fewer cells with disruption of the control *Ptprc* locus encoding CD45 (*Figure 1—figure supplement 2B*). Targeting $Klrc1_{5'E}$ increased the percentage of NKG2A[B6]-negative cells from ~10% to ~20%–40%. (*Figure 1—figure supplement 2C-E*), in line with our theoretical maximum editing efficiency. Similarly, targeting $Klra7_{Hss1}$ resulted in marked loss of Ly49G2[B6] expression, with minimal (<5%) loss of expression in non-targeting or non-nucleofected (no zap) control

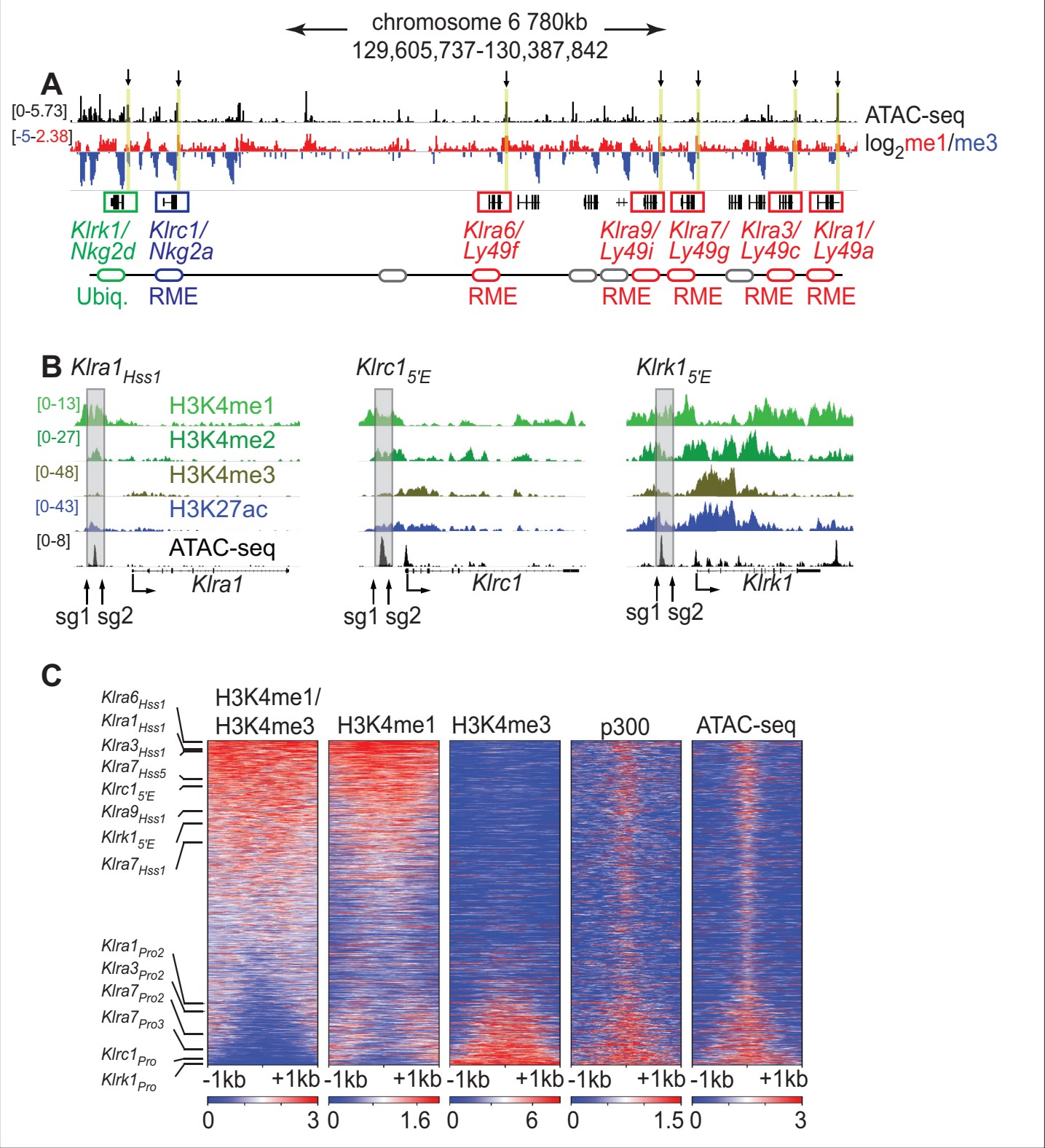

**Figure 1.** The *Klra1_Hss 1*, *Klrc1_5'E* and *Klrk1_5'E* elements display chromatin features of enhancers. (**A**) ATAC-seq and H3K4me1:me3 log₂ ratio ChIP-seq data of relevant NKC genes in primary NK cells; red denotes positive me1:me3 ratios (enhancer-like) while blue indicates negative values (promoter-like). Approximate gene locations are indicated (bottom). Standard gene names (*Klr* nomenclature) are indicated followed by names derived from the gene products (*Ly49* or *Nkg2*) for reference. Gray ovals represent additional undiscussed *Klra* genes. Vertical yellow bars and arrows denote the positions of the Hss1 and 5'E enhancers at the indicated genes. Data are sourced from ref (*Lara-Astiaso et al., 2014*). Normalized data ranges are indicated on the

*Figure 1 continued on next page*

*Figure 1 continued*

left. (**B**) Normalized ChIP-seq and ATAC-seq results (sourced from *Lara-Astiaso et al., 2014*), showing enhancer and promoter histone modifications at *Klra1*, *Klrc1*, and *Klrk1*. Approximate locations of sgRNAs used in this study to delete enhancers are shown. All datasets are presented with the same vertical scale across sub-panels, which are indicated in normalized signal per million reads (SPMR) in the left sub-panel. (**C**) Heatmaps depict 51,650 ATAC-seq peaks in primary NK cells (excluding peaks ranking in the bottom 5% for either H3K4me1 or H3K4me3) ranked according to H3K4me1:me3 ratio of average ChIP-seq signal calculated over a 2kb window centered on the ATAC-seq peak midpoint. The indicated data are displayed over these peaks in each heatmap. The locations of selected NK receptor gene Hss1, 5′E and promoter elements within the me1:me3 ranking are shown. H3K4 methylation data are sourced from ref (*Lara-Astiaso et al., 2014*) while p300 is sourced from ref (*Sciumè et al., 2020*).

The online version of this article includes the following figure supplement(s) for figure 1:

**Figure supplement 1.** Chromatin features and TF binding profile of the *Klra_{Hss1}*, *Klrc1_{5′E}*, and *Klrk1_{5′E}* enhancers.

**Figure supplement 2.** The *Klrc1_{5′E}* and *Klra7_{Hss1}* enhancers are required to maintain gene expression in primary NK cells.

conditions (*Figure 1—figure supplement 2F-H*). These data show that *Klrc1_{5′E}* and *Klra7_{Hss1}* play key roles in maintaining expression of active alleles in mature NK cells, arguing against the proposal that Hss1 elements are only required in immature NK cells (*Saleh et al., 2004*; *Saleh et al., 2002*). The results provide functional evidence that Hss1 functions as an enhancer in mature NK cells.

## The *Klra7_{Hss1}* and *Klrc1_{5′E}* enhancers are constitutively accessible

Analysis of bulk NK cells did not reveal a correlation between the gene expression frequency of an NK receptor gene and the accessibility, TF occupancy, or H3K27ac modifications of Hss1 and 5′E enhancers (*Figure 1—figure supplement 1A*). This lack of concordance raised the possibility that these enhancers were similarly active and occupied by TFs upstream of both silent and active alleles, as has been observed for RME genes genome-wide in F_1 clones (*Levin-Klein et al., 2017*; *Xu et al., 2017*). It has not previously been possible, however, to address this issue for an RME gene in freshly isolated ex vivo cell populations.

To purify populations of cells expressing different alleles of *Klrc1*, we stained (B6 x BALB/c)F_1 NK cells with allele-specific antibodies (*Vance et al., 2002*), allowing us to sort and perform ATAC-seq on NK cell populations expressing all four configurations of alleles: expressing both alleles of *Klrc1*, only B6, only BALB/c, or neither (*Figure 2A and B*). While the cells expressing both alleles and those expressing only the B6 allele are closely juxtaposed in the cytometry plots, post sort analysis demonstrated that they could be efficiently separated (*Figure 2—figure supplement 1A*), consistent with previously published data where allelic expression was confirmed by mRNA analysis (*Rogers et al., 2006*). SNPsplit (*Krueger and Andrews, 2016*) analysis of reads demonstrated that the enhancer element *Klrc1_{5′E}* was accessible on both active and inactive alleles in all four populations, whereas the *Klrc1* promoter was accessible only at active alleles (*Figure 2B*). We used a similar allele-specific staining protocol (*Tanamachi et al., 2001*) to sort and analyze cells expressing either, both or neither Ly49G2 allele (*Figure 2C, D*). The *Klra7_{Hss1}* enhancer was accessible on both active and inactive alleles in all four populations, whereas the dominant promoter Pro3 (*Gays et al., 2011*) was accessible only on the active allele (*Figure 2D*). Notably, the *Klra7* gene harbors a second minor enhancer element, *Klra7_{Hss5}* (*Figure 1C*; *Figure 1—figure supplement 1B*), which was similarly accessible at all alleles (*Figure 2D*).

These data demonstrated that enhancers within the *Klra* and *Klrc* gene families behave similarly to those of other RME genes analyzed in F_1 hybrid clones (*Levin-Klein et al., 2017*; *Xu et al., 2017*), exhibiting an accessible configuration whether or not the gene was expressed. Importantly, this analysis further validated the NK receptor genes as a model for RME. While initially surprising, the decoupling of enhancer and promoter accessibility seen at NK receptor genes and other RME loci is consistent with a binary model of enhancer action, where enhancer activation occurs in all cells of a given type and acts stochastically to raise the binary "on or off" probability of gene expression, rather than regulate the per-cell amount of expression (*Blackwood and Kadonaga, 1998*; *Walters et al., 1995*).

We extended these observations by analyzing the pattern of active enhancer associated marks at silent and active alleles of *Klra7* (which encodes Ly49G2). In order to obtain sufficient cell numbers of rare populations, we expanded NK cells from (B6 x BALB)F_1 mice with IL-2 containing medium and then sorted cells that expressed neither (N) or both (B) Ly49G2 alleles and performed CUT&RUN for the enhancer-associated H3K4me1/2/3 and H3K27ac modifications. The *Klra7* promoter and

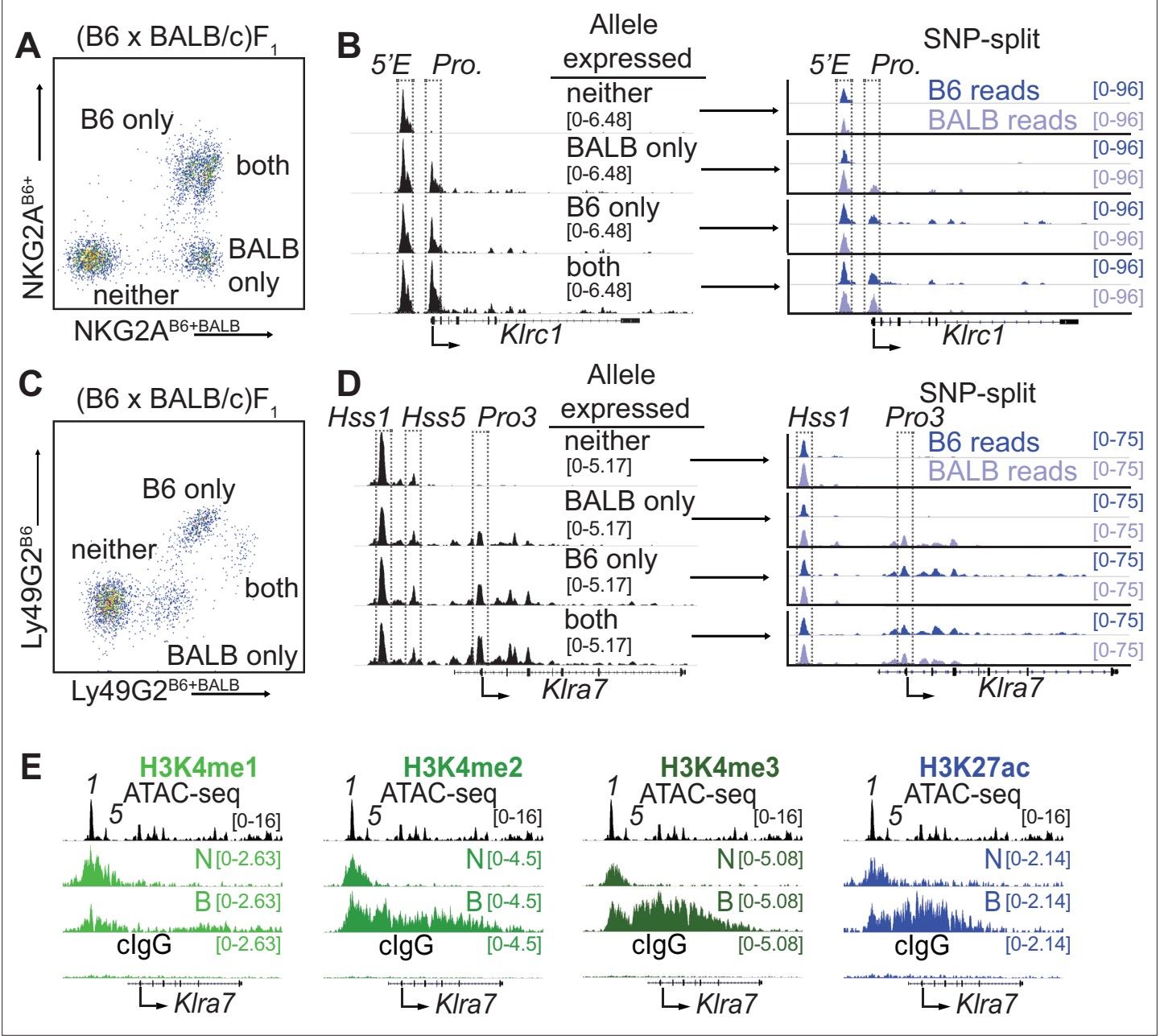

**Figure 2.** *Klrc1₅'E* and *Klra7_Hss1* are constitutively accessible, while promoters are accessible only at expressed alleles. (**A**) FACS plot depicting splenic NK cells from a (B6 x BALB/c)F₁ hybrid mouse stained with allele-specific antibodies, allowing separation of NK cells expressing both, either, or neither NKG2A allele. (**B**) (left) Normalized ATAC-seq data generated from the 4 cell populations depicted in (**A**) aligned to the mm10 reference genome. (right) Allele-informative reads were binned according to chromosome of origin, and displayed as signal mapping to the B6 or BALB/c chromosome. The *Klrc1₅'E* enhancer and promoter (*Pro.*) are boxed (dotted line). Vertical data range in SPMR is indicated for each track. (**C and D**) Data are as in (**A and B**), but using an allele-specific staining protocol with respect to the Ly49G2 receptor. *Klra7_Hss1*, *Klra7_Hss5* and the dominant TSS (*Pro3*, *Gays et al., 2011*) are boxed. (**E**) CUT&RUN data depicting each of 4 indicated histone modifications at the *Klra7* gene in IL-2 expanded NK cells sorted to express neither 'N' or both 'B' alleles of *Klra7*. Negative control CUT&RUN data were generated using a mouse IgG2a κ (cIgG) antibody, and a 50:50 mixture of IL-2 expanded NK cells that expressed the B6 or BALB/c alleles. These data are displayed in the bottom track in each sub-panel. The ATAC-seq patterns are shown for reference above each analysis; *Klra7_Hss1* is denoted as '1', *Klra7_Hss5* is denoted as '5'. Arrows depict the locations of the dominant *Pro3* TSS. All ATAC-seq and CUT&RUN data within a sub-panel are presented with the same vertical scale.

The online version of this article includes the following figure supplement(s) for figure 2:

**Figure supplement 1.** Post-sort analysis of sorted NK cell populations used for ATAC-seq and CUT&RUN analysis.

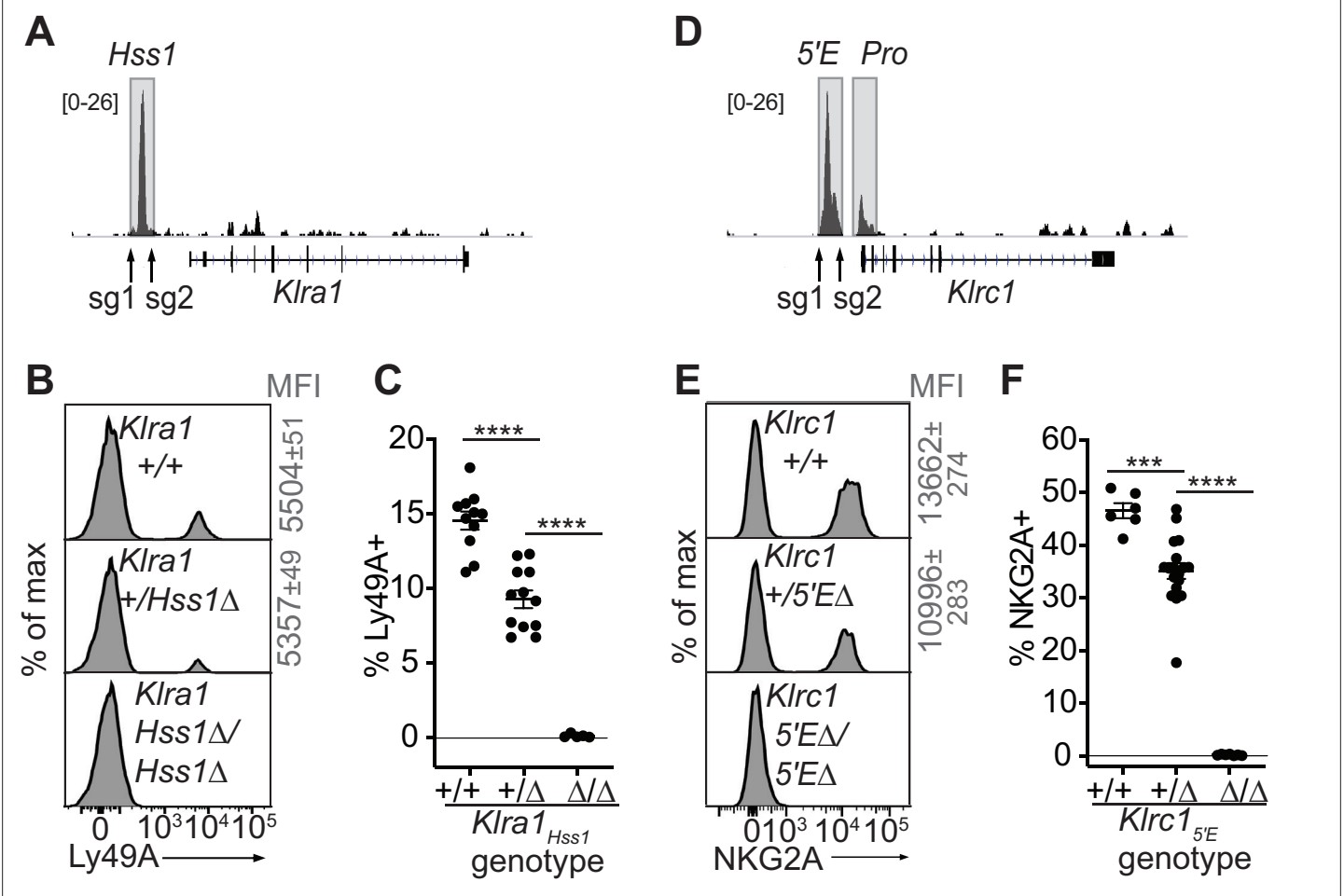

**Figure 3.** The *Klra1_Hss1* and *Klrc1_5'E* enhancers are required for gene expression. (**A**) Locations of sgRNAs used to delete *Klra1_Hss1* in the B6 germline; NK cell ATAC-seq are displayed for reference with the vertical data scale in SPMR indicated. (**B–C**) Ly49A staining of the indicated *Klra1_Hss1* deletion littermates. MFI of staining ± SEM are depicted in gray. In (**C**), data are combined from two independent experiments (means ± SEMs, n = 5–12). (**D–F**) Data as in (**A–C**) for *Klrc1_5'E*. Data in (**F**) are combined from two experiments with the *Klrc1_5'E*(B3Δ) allele (*Figure 3—figure supplement 1C*) and were recapitulated in analysis of the B1Δ allele (means ± SEMs, n = 6–18). ****p < 0.0001; ***p < 0.0001 using one-way ANOVAs with Tukey's multiple comparisons.

The online version of this article includes the following figure supplement(s) for figure 3:

**Figure supplement 1.** *Klra1_Hss1Δ* and *Klrc1_5'EΔ* alleles employed in the study.

**Figure supplement 2.** The constitutively accessible *Klrc1_5'E* and *Klra1_Hss1* enhancers act entirely in *cis*.

gene body displayed striking enrichment of H3K4me2/3 and H3K27ac in cells that expressed both *Klra7* alleles, and as predicted lacked these modifications in cells where neither allele was expressed (*Figure 2E*). Notably, the *Klra7_Hss1* and *Klra7_Hss5* enhancers displayed equal enrichment of H3K27ac in cells expressing both alleles or neither (*Figure 2E*). As H3K27ac delineates active as opposed to poised enhancers (*Calo and Wysocka, 2013*), these data suggest constitutive enhancer activation on both silent and active alleles. These data are consistent with previous results demonstrating that enhancer-derived transcripts are produced at Hss1 elements in cells that do not express the target *Klra* gene (*Gays et al., 2015*). Thus, whether measured by accessibility, H3K27ac enrichment or eRNA production, Hss1 activity may be decoupled from target *Klra* expression status.

### *Klra1_Hss1* and *Klrc1_5'E* are required for gene expression in vivo, and act in *cis*

We tested the requirement for *Klra1_Hss1* in the endogenous locus by deleting it in the B6 germline via CRISPR/Cas9 editing (*Figure 3A-C*, *Figure 3—figure supplement 1A, B*). *Klra1^Hss1Δ/Hss1Δ* mice

completely lacked Ly49A expression (*Figure 3B, C*; *Figure 3—figure supplement 1B*), but importantly expression of other Ly49 receptors was unaffected (*Figure 3—figure supplement 1B*), supporting the notion that the variegated NK receptor genes are regulated proximally and independently of each other. *Klra1$^{+/Hss1\Delta}$* mice displayed an intermediate percentage of Ly49A+ cells (*Figure 3B and C*; *Figure 3—figure supplement 1B*).

As with *Klra1$_{Hss1}$*, deletion of both allelic copies of *Klrc1$_{5'E}$* in the germline eliminated NKG2A expression, and heterozygous mice displayed a reduced frequency of NKG2A+ NK cells (*Figure 3D-F*; *Figure 3—figure supplement 1C, D*).

Whether the activity of constitutively accessible enhancers of RME genes is coordinated in *trans* via a dedicated epigenetic mechanism that enacts RME is not known. We addressed the *cis* vs *trans* activity of *Klrc1$_{5'E}$* and *Klra1$_{Hss1}$* in F$_1$ hybrids using allele-discriminating antibodies. F$_1$ hybrid mice between B6-*Klrc1$_{5'E\Delta}$* mice and BALB/c mice were generated (*Klrc1$^{B6-5'E\Delta/BALB/c+}$* heterozygotes) along with WT F$_1$ littermates. The percentages of cells expressing all four combinations of NKG2A alleles were determined using allele-specific NKG2A antibodies we previously generated. Assuming that the *Klrc1$_{5'E}$* acts only in *cis* we calculated the expected changes in the frequencies of these cells in the heterozygotes. The experimental data closely mirrored the predictions (*Figure 3—figure supplement 2A-C*). Therefore, the constitutively accessible *Klrc1$_{5'E}$* acted in *cis* and independently of the activity of the other copy. Similarly, in *Klra1$^{B6-Hss1\Delta/BALB/c+}$* heterozygotes, the BALB/c allele was unaffected when expression of the B6 allele was abrogated (*Figure 3—figure supplement 2D-F*).

## A *cis*-acting secondary enhancer in the *Klra7* locus contributes to the high expression frequency of *Klra7*

Both the *Klra1* and *Klrc1* gene loci harbor only a single prominent proximal enhancer-like site (*Figure 1A and B*), and are completely dependent on those enhancers for expression (*Figure 3*), complicating analysis of the role of *Klra1$_{Hss1}$* and *Klrc1$_{5'E}$* in the RME of their target genes. We reasoned that analysis of an RME NK receptor gene with multiple proximal enhancers could reveal the role of overall enhancer strength in regulating expression frequency. We hypothesized, in accordance with the binary model, that despite the presence of multiple (weak) enhancers, enhancer activity at such loci is limiting resulting in RME. We predicted that limiting enhancer activity further by deleting a secondary enhancer in a natural RME gene would reduce, but not abrogate, gene expression probability.

The *Klra7* locus is expressed by ~50% of NK cells and contains both an *Hss1* element and another constitutively accessible enhancer, *Klra7$_{Hss5}$* (*Figure 2D*). Interestingly, the corresponding region of the highly related *Klra1* gene, which is expressed by only ~17% of NK cells, is much less accessible and presumably less active (*Figure 4A*).

Germline deletion of *Klra7$_{Hss5}$*, in homozygous configuration, resulted in a depressed percentage of Ly49G2+ cells (35%) compared to WT mice (50%), with only a minor change in expression per cell (measured by mean fluorescence intensity of staining, MFI) (*Figure 4B-D*; *Figure 4—figure supplement 1A and B*). Heterozygous mice displayed an intermediate percentage of Ly49G2+ cells (*Figure 4B, C*; *Figure 4—figure supplement 1B*). To test whether *Klra7$_{Hss5}$* acts entirely in *cis*, we crossed *Klra7$_{Hss5\Delta}$* to BALB/c mice. The NK cell populations in the F$_1$ mice expressing the Ly49G2$^{B6}$ alleles were reduced, and the populations expressing neither allele or only Ly49G2$^{BALB/c}$ were increased, in the proportion expected under probabilistic action of *Klra7$_{Hss5}$* in *cis* (*Figure 4E, F*). Thus, the constitutively active enhancer *Klra7$_{Hss5}$* is directly involved in regulating Ly49G2 expression frequency, and explains, at least in part, the high expression frequency of Ly49G2 in relation to other receptors including Ly49A.

## Deletion of *Klrk1$_{5'E}$* is sufficient to recapitulate stable RME in *Klrk1*

Our hypothesis that RME of NK receptor genes is imparted by limiting binary enhancer activity predicts that a receptor expressed by all NK cells may be converted into a variegated receptor by weakening enhancer activity, for example by deleting one of multiple associated enhancer elements. We tested this for the *Klrk1* gene encoding the NKG2D immunostimulatory receptor, which is expressed by all NK cells (*Wensveen et al., 2018*), is distantly related to the *Klrc1* and *Klra* genes, and is flanked on both sides by enhancer-like chromatin, suggesting possible regulation by multiple enhancers (*Figure 1B*). Deletion of the enhancer-like ATAC-accessible site ~5 kb upstream of the *Klrk1* gene (*Klrk1$_{5'E}$*) (*Figure 1B*; *Figure 5—figure supplement 1A, B*), resulted in variegated NKG2D expression in *Klrk1$^{5'E\Delta/5'E\Delta}$* animals (*Figure 5A and B*). Only ~65% of NK cells expressed NKG2D in *Klrk1$^{5'E\Delta/5'E\Delta}$*

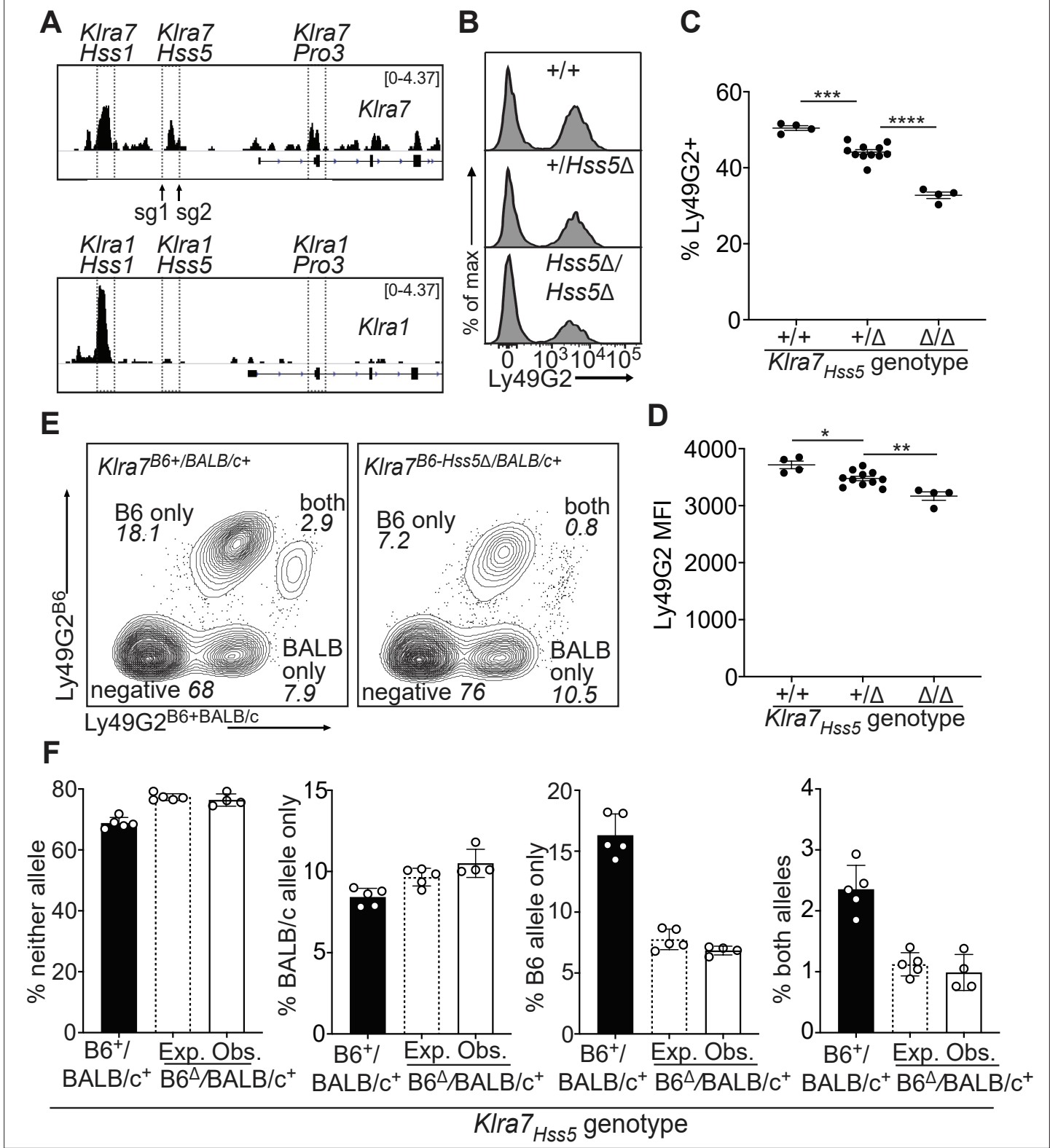

**Figure 4.** A minor *cis*-acting enhancer amplifies Ly49G2 expression frequency. (**A**) Normalized ATAC-seq tracks of *Klra1* and *Klra7* in bulk NK cells; the vertical data range in SPMR is displayed on each track. Hss1 and Hss5 enhancers and the Pro3 promoter are highlighted. sgRNAs used to generate *Klra7$_{Hss5\Delta}$* alleles are shown (arrows). (**B**) Ly49G2 staining of NK cells in the indicated *Klra7$_{Hss5}$* deletion littermates (B2Δ allele, *Figure 4—figure supplement 1A*). (**C–D**) Ly49G2 percentages (**C**), and mean fluorescence intensities of the positive populations (**D**) (n = 4–11). Similar results were obtained with the B1Δ allele (*Figure 4—figure supplement 1B*). (*p < 0.05; **p < 0.01; ***p < 0.001; ****p < 0.0001 using One-way ANOVA with Tukey's

*Figure 4 continued on next page*

*Figure 4 continued*

multiple comparisons). (**E**) Flow cytometry plots of gated *Klra7^B6-Hss5Δ/BALB/c+* NK cells using Ly49G2^B6-specific and Ly49G2^B6+BALB/c-specific antibodies (right) and a wildtype littermate (left). (**F**) Expected and observed percentages of populations depicted in 'E' in F1 mice with the *Klra7_Hss5Δ* (hatched bar is expected, white bar is observed) or wildtype (black) *Klra7* allele. Expected frequencies were calculated assuming stochastic *cis* regulation of alleles (see Materials and methods; note effect of genetic background). Data are representative of two experiments. All error bars represent SEM.

The online version of this article includes the following figure supplement(s) for figure 4:

**Figure supplement 1.** *Klra7_Hss5Δ* alleles employed in this study.

animals. The expression level per cell was only modestly affected and to an extent consistent with largely monoallelic expression vs expression in some cells of both alleles (***Figure 5C*** and ***Figure 5—figure supplement 1C***). These data suggest that the primary role of *Klrk1_{5'E}* is to increase the probability rather than the degree of *Klrk1* expression, in line with the binary model of enhancer action.

Significantly, the expression state of *Klrk1_{5'EΔ}* alleles was mitotically stable. NK cells from the enhancer knockouts were stimulated with IL-2 for 2–3 days before sorting NKG2D+ and NKG2D- populations, and further expanded in IL-2 for an additional 8–10 days, where they underwent an ~10–100 fold expansion. The NKG2D+ and NKG2D- phenotypes were highly stable despite extensive proliferation (***Figure 5D***).

In heterozygotes with the *Klrk1_{5'EΔ}* allele on one chromosome and a *Klrk1* exon replacement knockout allele (***Guerra et al., 2008***) on the other (-/5'EΔ) the percentage of NKG2D+ cells was lower than in 5'EΔ/5'EΔ mice (***Figure 5E and F***). This nearly matched the expected percentage under the assumption that the 5'EΔ alleles are independently regulated in the heterozygotes, that is, the positive cells include cells expressing both alleles with a frequency that is the product of the individual frequencies (***Raulet et al., 1997***; ***Figure 5G***). NKG2D expression per NKG2D+ cell in *Klrk1^{5'EΔ /5'EΔ}* animals appeared slightly higher than in *Klrk1^{+/-}*, consistent with a proportion of cells expressing both *Klrk1* alleles, a feature characteristic of natural RME (***Eckersley-Maslin and Spector, 2014***; ***Gendrel et al., 2016***; ***Figure 5E*** and ***Figure 5—figure supplement 1C***). Together, these data strongly argue that expression of *Klrk1_{5'EΔ}* alleles follows a stochastic RME pattern.

## The *Klrk1_{5'EΔ}* allele mimics the expression and accessibility features of naturally variegated NK receptor genes

The stable RME of *Klrk1_{5'EΔ}* alleles recapitulated the stochastic expression pattern of naturally variegated NK receptor genes. Expression of NKG2D in *Klrk1^{5'EΔ/5'EΔ}* mice was approximately randomly distributed with respect to the expression of the RME genes encoding NKG2A, Ly49G2 or Ly49I (***Figure 5H***). Indeed, the co-expression frequencies approximated the products of the separate frequencies of the receptors studied (the 'product rule' ***Raulet et al., 1997***; ***Figure 5I***).

To examine the chromatin accessibility of the *Klrk1* locus in *Klrk1^{5'EΔ/5'EΔ}* mice, we performed ATAC-seq with NKG2D+ and NKG2D- cells sorted from *Klrk1^{5'EΔ/5'EΔ}* mice (***Figure 5J***). Robust promoter accessibility was detected in NKG2D+ cells but not in NKG2D- cells (***Figure 5K***). The 5'E element was deleted in both populations and therefore not accessible, but a 3' enhancer-like element was equally accessible in both populations. This accessibility pattern mirrors that of the naturally variegated NK receptor genes (***Figure 2***) and RME broadly (***Xu et al., 2017***). We speculate that this 3' element works in concert with 5'E in WT cells to increase the expression frequency of *Klrk1* in NK cells, and represents the residual enhancer that drives variegated *Klrk1* expression after 5'E deletion. These speculations remain to be tested.

The results of our experiments with the *Klrk1* gene established that stable RME and the stochastic and variegated NK receptor expression pattern could be recapitulated in full by weakening enhancer activity at a gene normally expressed in ~all NK cells. Furthermore, the similarity of *Klrk1^{5'EΔ/5'EΔ}* and natural NK receptor gene variegation suggests that previous examples of enhancer deletion-associated variegation such as that seen in the *Cd8a* locus (***Ellmeier et al., 2002***; ***Garefalaki et al., 2002***) are rooted in similar mechanisms as naturally-occuring RME.

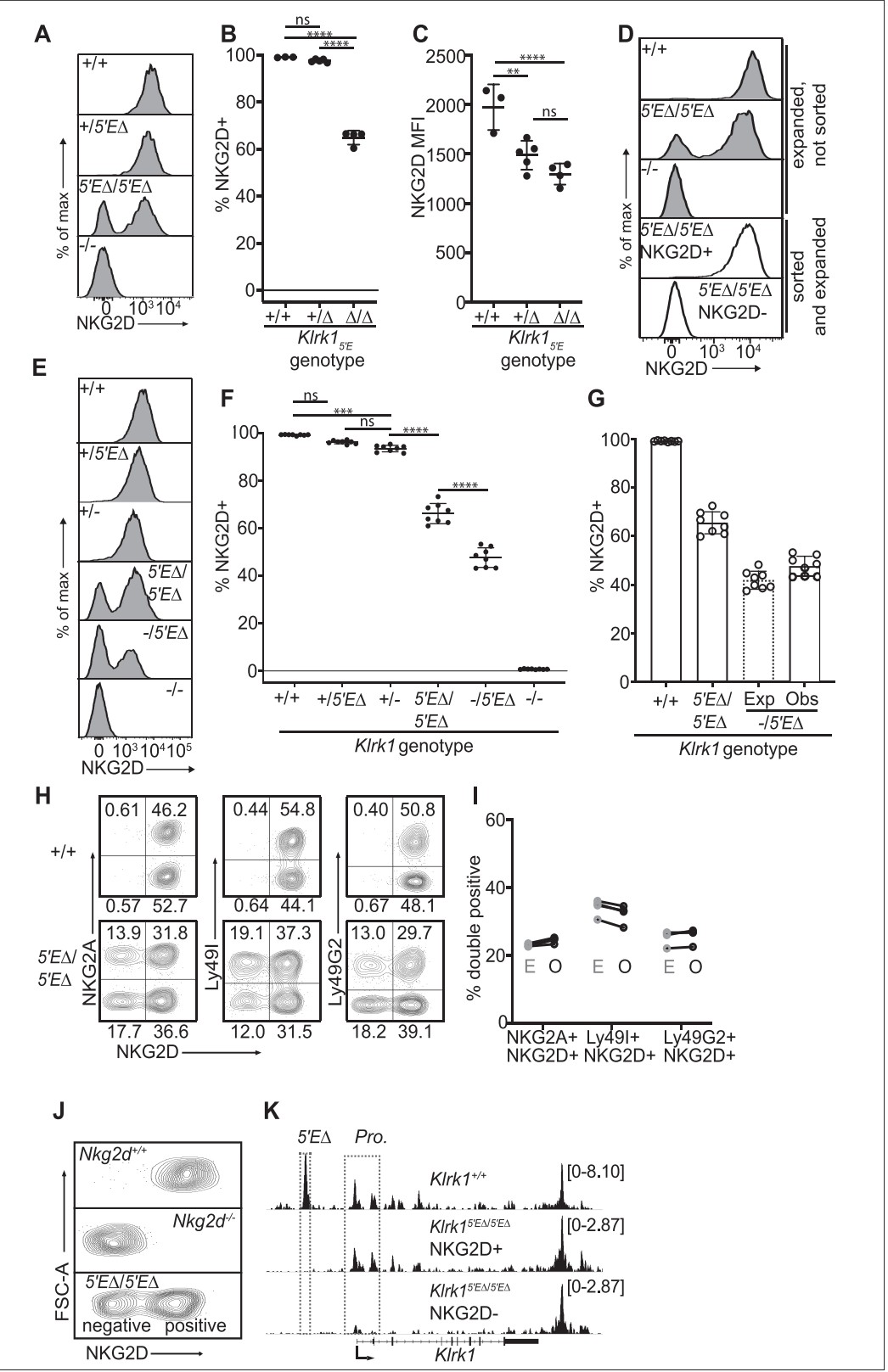

**Figure 5.** *Klrk1_{5'E}* deletion results in mitotically stable RME, fully recapitulating natural variegation. (**A–C**) NKG2D staining of splenocytes from *Klrk1_{5'E}* deletion littermates (B1Δ allele), and an *Klrk1^{-/-}* mouse (n = 3–5). Results are representative of four experiments with two deletion alleles (*Figure 4—figure supplement 1*, **C and D**). (**D**) Splenocytes from *Klrk1^{5'EΔ/5'EΔ}* mice were cultured with IL-2 for 2–3 days before sorting NKG2D+ and NKG2D- NK

*Figure 5 continued on next page*

*Figure 5 continued*

cells, which were expanded in fresh IL-2 medium for 8–10 days before analysis (white fill). Expanded, unsorted NK cells are shown in gray. (**E**) Staining of splenic NK cells from mice of six genotypes. "+", "-" and "Δ" refer to wildtype, gene knockout, and *Klrk1$_{5'E}$* deletion alleles, respectively. (**F**) Quantified results in (**E**) compiled from two experiments. (**G**) Expected and observed percentages of NKG2D+ NKcells in *Klrk1$^{-/5'EΔ}$* mice. Expected expression is calculated based on observed NKG2D+ percentages in *Klrk1$^{5'EΔ/5'EΔ}$* mice, assuming stochastic expression (see Materials and methods). Data are comprised of selected groups displayed in (**G**). (**H**) Stochastic co-expression of NKG2D and NKG2A, Ly49I or Ly49G by NKp46+ NKcells in *Klrk1$^{5'EΔ/5'EΔ}$* mice. WT (+/+) mice are shown for comparison. (**I**) Expected ('E') and observed ('O') percentages of cells coexpressing the indicated receptors in *Klrk1$^{5'EΔ/5'EΔ}$* mice. Expected percentages were calculated by mutiplying percentages of cells in each mouse expressing each receptor individually (n = 4). Data are representative of two experiments. (**J**) NKG2D staining of presorted gated NK cells from *Klrk1$^{5'EΔ/5'EΔ}$* mice (bottom), compared to wildtype and *Klrk1$^{-/-}$* NK cells. (**K**) Normalized ATAC-seq tracks generated from NKG2D+ and NKG2D- cells sorted from the *Klrk1$^{5'EΔ/5'EΔ}$* mouse shown in (**J**) and are presented on the same vertical scale. ATAC-seq results for WT splenic NK cells were sourced from *Lara-Astiaso et al., 2014* and auto-scaled to match the data generated from the *Klrk1$^{5'EΔ/5'EΔ}$* mouse. Vertical data scale in SPMR is displayed for each track. Error bars represent SEM. **p < 0.01; ***p < 0.001; ****p < 0.0001, computed using One-way ANOVAs with Tukey's multiple comparisons.

The online version of this article includes the following figure supplement(s) for figure 5:

**Figure supplement 1.** *Klrk1$_{5'Δ}$* alleles employed in this study.

**Figure supplement 2.** The *Klrk1* promoter is CpG methylated at silent alleles in *Klrk1$^{5'EΔ/5'EΔ}$* NK cells.

## Silent *Klrk1* alleles in *Klrk1$^{5'EΔ/5'EΔ}$* NK cells are CpG methylated at the promoter

Bisulfite conversion, PCR amplification, cloning and Sanger sequencing were carried out to examine methylation of the promoter region of *Klrk1* in sorted NKG2D+ and NKG2D- NK cells from *Klrk1$^{5'EΔ/5'EΔ}$* mice, or similar NK cell populations that had been expanded in IL-2-containing medium for 8 days. Silent alleles in NKG2D-negative NK cells from *Klrk1$^{5'EΔ/5'EΔ}$* mice were hypermethylated at five CpG sites in an ~150 bp stretch surrounding the promoter, while these CpG sites were largely unmethylated in NKG2D+ NK cells, whether the latter cells came from *Klrk1$^{5'EΔ/5'EΔ}$* mice or WT mice (*Figure 5— figure supplement 2A, B*).

Correlations between expression and the absence of methylation of the corresponding promoters of the *Klra* and *Klrc1* genes have been previously documented (*Rouhi et al., 2007*; *Rouhi et al., 2006*; *Rouhi et al., 2009*). The methylation of these promoters, or of *Klrk1* promoters as documented here, is not necessarily a sufficient cause of silenced gene expression, however. For example, DNA methylation of promoters is not associated with silent alleles in the case of numerous other RME genes, nor is it necessary for maintaining the silenced state of the large majority of tested RME genes (*da Rocha and Gendrel, 2019*; *Eckersley-Maslin et al., 2014*; *Gupta et al., 2022*; *Marion-Poll et al., 2021*). In addition, inhibitors of DNA methyltransferases did not result in activation of silenced *Klra* genes or the *Klrc1* gene (*Rouhi et al., 2007*; *Rouhi et al., 2006*; *Rouhi et al., 2009*). Treatments with 5-azacytidine proved to be toxic to primary NK cells, which precluded our ability to test the role of methylation in the maintenance of *Klrk1* silencing.

A notable finding was that the same five CpG sites in the *Klrk1* promoter were usually methylated in hematopoietic stem cells, which are progenitors of mature NK cells. Specifically, 13/16 of these sites were methylated based on analysis of whole genome bisulfite sequencing (WGBS) reads (*Li et al., 2021*). The promoter of the *Klrc1* gene was also largely methylated in hematopoietic stem cells (*Rogers et al., 2006*), and we extended this analysis to the *Klra1* promoter and *Klra7* promoters. Using the dataset generated in *Li et al., 2021*, we found that nearly all the CpGs analyzed in 500 bp windows surrounding each promoter were methylated in hematopoietic stem cells (18 of 18 CpGs in reads mapping to *Klra1*, and 15/19 CpGs in reads mapping to *Klra7*). These findings suggest that activation of RME alleles is associated with demethylation, as opposed to a model where unmethylated promoters of RME genes are selectively methylated as a means to impose RME.

## Silent NK receptor gene alleles lack repressive histone modifications associated with polycomb and heterochromatic repression

We investigated histone modifications associated with gene repression to search for clues regarding the maintenance of the active and silent epigenetic states. We assayed the polycomb-associated marks H3K27me3 and H2AK119Ub1 (H2AUb1) and the heterochromatin-associated H3K9me3, which have previously been found at inactive alleles of some other monoallelically expressed genes, notably the odorant receptors and protocadherins (*Eckersley-Maslin and Spector, 2014*; *Eckersley-Maslin et al., 2014*; *Gendrel et al., 2014*; *Gendrel et al., 2016*; *Xu et al., 2017*). CUT&RUN analysis of repressive modifications in IL-2 expanded primary NK cells that were sorted as Ly49G2-negative (expressing neither allele, designated 'N'), revealed that all three modifications were prevalent in the *Hoxa* gene cluster, as expected, but the entire NKC lacked appreciable signal for any of the modifications (*Figure 6A*). None of the 3 marks were enriched above background on silent *Klra7* alleles (*Figure 6B*). Importantly, many other genes associated with non-NK cell lineages (e.g. *Cd19* and *Mstn*, expressed in B cells and myocytes, respectively) were also not enriched for these repressive modifications (*Figure 6B*). In contrast, other genes such as *Pdcd1* (encodes PD-1) and *Spi1* (encodes the macrophage and B-cell lineage-regulating transcription factor PU.1) displayed all 3 marks. Therefore, with respect to repressive chromatin marks, silent *Klra7* alleles resembled several genes normally expressed in other hematopoietic cell lineages but not in NK cells, rather than known repressed genes.

As silent *Klra7* alleles appeared similar to lineage non-specific genes in our analysis of repressive chromatin marks, we extended the analysis of chromatin states using ChromHMM, which integrates multiple datasets to classify the genome into subdomains based on their chromatin signatures (*Ernst and Kellis, 2012*). Using data from cells expressing neither or both Ly49G2 alleles, we constructed a minimal 3 state model corresponding to transcriptionally active chromatin (high levels of H3K27ac and H3K4me3, both active marks), repressed chromatin (H2AUb1 and H3K9me3, both repressive marks), and inactive chromatin (lacking these active or repressive marks) (*Figure 6—figure supplement 1A*). As expected, the promoters of lineage-appropriate genes expressed in NK cells (e.g. *Ncr1*, *Klrb1c*, *Ifng*) fell into the 'active' chromatin state 1 (*Figure 6—figure supplement 1B*). Notably, genes commonly regarded as markers of non-NK cell hematopoietic lineages (e.g. *Cd3e*, *Cd19*, *Ly6g*, *Siglech*) fell into the 'inactive' chromatin state 2 (*Figure 6—figure supplement 1C*). Finally, promoters of other genes, often encoding transcription factors that promote non-NK cell fates such as *Bcl11b*, *Batf3*, and *Pax5*, fell into the 'repressed' state 3 (*Figure 6—figure supplement 1D*). These data suggest that many genes encoding immune effector molecules associated with non-NK lineages are not actively repressed but are inactive and stably silent, whereas genes promoting non-NK cell fates are actively repressed.

In cells expressing both copies of *Klra7*, the enhancer, promoter and gene body all fell within the active state 1 (*Figure 6—figure supplement 1E*), whereas in cells expressing neither copy, the enhancer remained in the active state 1 but the promoter and gene body became inactive (state 2) rather than repressed (state 3). Indeed, it was striking that the NKC as a whole lacked repressive state 3 chromatin (*Figure 6A*; *Figure 6—figure supplement 1E*). The lack of the repressive chromatin state at silent NK receptor genes suggests that repressive chromatin may not be required for stable RME generally, potentially explaining why repressive chromatin signatures are not a consistent feature of silent RME alleles in other instances (*Eckersley-Maslin and Spector, 2014*; *Eckersley-Maslin et al., 2014*; *Gendrel et al., 2014*; *Gendrel et al., 2016*). In lieu of active repression, other mechanisms must be invoked for the maintenance of RME patterns through cell division.

## The lineage-defining receptor genes *Ptprc/Cd45*, *Klrk1*, *Cd8a* and *Thy1* are monoallelically expressed, suggesting RME may be ubiquitous

Our findings that RME is rooted in broad and probabilistic properties of gene activation raised the possibility that these principles might apply to many and perhaps all genes. We hypothesized that many genes exhibit a minor extent of RME but escaped previous detection due to the methods used, which were limited by clone numbers and therefore lacked the resolution to detect very rare monoallelic expression (*Eckersley-Maslin et al., 2014*; *Gendrel et al., 2014*; *Gimelbrant et al., 2007*; *Reinius et al., 2016*). We employed flow cytometry to analyze millions of primary cells ex vivo for rare monoallelic expression of several membrane proteins, starting with NKG2D. We noticed that ~2.5%

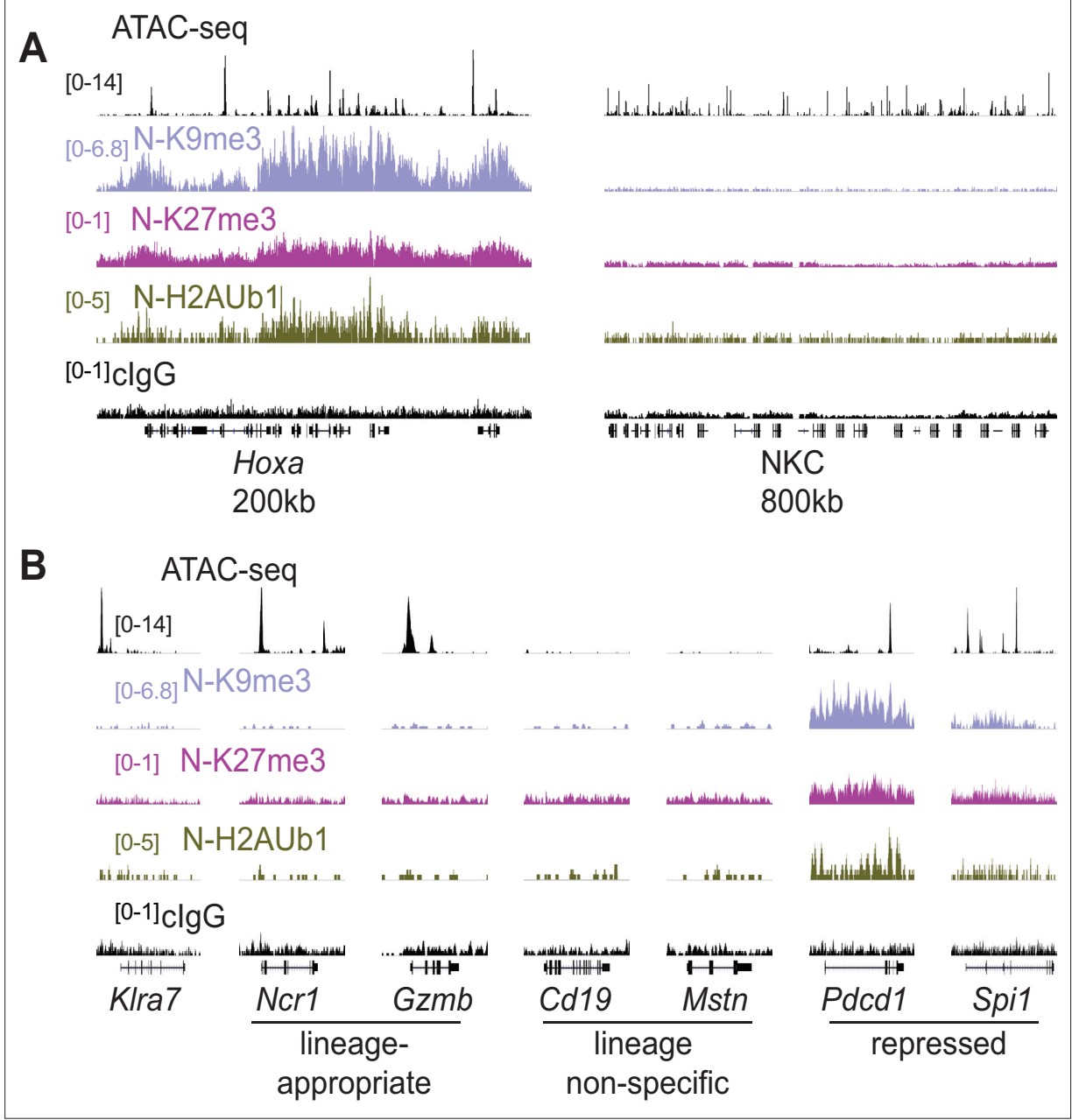

**Figure 6.** Silent NK receptor gene alleles resemble inactive genes expressed in non-NK lineages, rather than repressed genes. (**A**) Repressive histone modification CUT&RUN data generated with primary IL-2 expanded NK cells sorted to express neither allele of Ly49G2 ('N' cells). IGV screenshots depicting the indicated histone modification or analyses with control mouse IgG2a κ (cIgG), which binds protein A. The *Hoxa* gene cluster (left) serves as a positive control. The entire NKC gene cluster is displayed on the right. The vertical scales in SPMR, indicated on the left of the panels, were matched for each type of mark for all samples analyzed and were chosen to provide strong signals for the positive control *Hoxa* cluster. The cIgG data were scaled the same as the H3K27me3 data, which had the weakest signal of the marks analyzed. (**B**) Data are displayed as in (**A**), at *Klra7* (left), and gene loci belonging to the following classes: NK cell lineage-appropriate, NK cell lineage non-specific, and loci repressed in NK cells.

The online version of this article includes the following figure supplement(s) for figure 6:

**Figure supplement 1.** Chromatin state analysis of NK cells expressing neither (N) allele or both (B) alleles of Ly49G2.

of NK cells in *Klrk1*$^{+/-}$ mice lacked expression of NKG2D, whereas the percentage in *Klrk1*$^{+/+}$ mice was close to 0% (*Figure 5F*; *Figure 7A, B*). These data suggested that rare monoallelic expression of WT *Klrk1* was obscured by expression of at least one allele in nearly all NK cells. Indeed, assuming allelic independence, the failure of each allele to be expressed in a random 2.5% of all NK cells (from

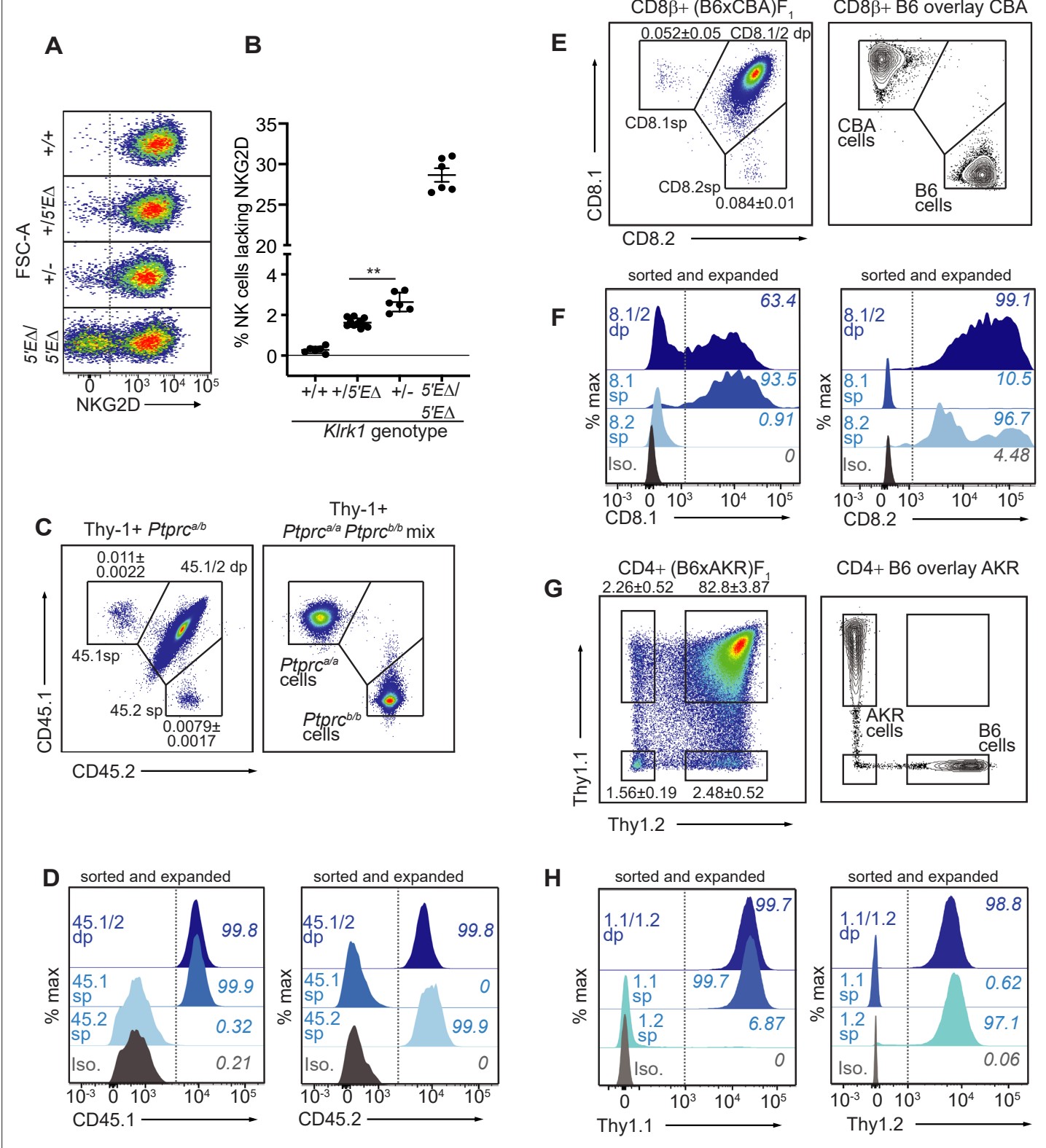

**Figure 7.** The lineage-defining *Klrk1, Ptprc, Cd8a* and *Thy1* genes are RME genes. (**A, B**) Flow cytometry (**A**) and quantification of % NKG2D-negative cells (**B**) of selected *Klrk1* genotypes. p = 0.0021, student's *t*-test. (**C**) Monoallelic CD45 expression. Flow cytometry of gated Thy-1+ cells pooled from 2 *Ptprc^{a/b}* mice (left). The mean percentages ± SEM of each monoallelic population, combined from three experiments, are depicted within the plot. Right panel: a mixture of cells from *Ptprc^{a/a}* and *Ptprc^{b/b}* mice. (**D**) CD45 allele single positive and double positive T cell populations were sorted from

*Figure 7 continued on next page*

*Figure 7 continued*

*Ptprc^{a/b}* mice using gates in panel C, expanded for 1 week in vitro, resorted to purity and expanded an additional ~5–8 fold. Histograms show CD45.1 and CD45.2 staining for the sorted populations after expansion. (**E–F**) Monoallelic expression of CD8α in (B6 x CBA)F1 mice presented as in (**C**) and (**D**). (**F**) shows CD8β+ cells from F₁ mice sorted and expanded twice as in (**D**) (~5–10 fold expansion in the second stumulation). (**G–H**) Data are displayed as in (**C–F**) but with respect to Thy-1 allelic expression on CD3+CD4+ T cells in (B6 x AKR)F₁ hybrid mice (6–8 fold expansion in the second stimulation). All experiments are representative of 2–3 performed. Error bars represent SEM.

The online version of this article includes the following figure supplement(s) for figure 7:

**Figure supplement 1.** Similar patterns of NKG2A, Ly49I and Ly49G2 expression in NKG2D+ and NKG2D- NK cells.

**Figure supplement 2.** Monoallelic expression of receptors thought to be expressed by all cells in various hematopoietic lineages compared to NK cell receptors.

here on referred to as an 'allelic failure rate') translated to only 0.063% of cells lacking both alleles. Importantly, in both *Klrk1^{5'EΔ/5'EΔ}* and *Klrk1^{+/-}* mice, NKG2D-negative cells were as likely as NKG2D+ cells to express NKG2A, Ly49G2, or Ly49I (**Figure 7—figure supplement 1A-C**). These data suggest that in both 5'E knockout and WT mice, NKG2D is randomly expressed with respect to the other RME receptor genes. Thus, even the WT *Klrk1* gene is expressed in an RME fashion.

To extend this approach, we sought to analyze allelic expression of 'lineage-defining' receptor genes for which (A) allele-specific antibodies exist and, (B) expression is normally considered to be universal in defined lymphohematopoietic lineages. We first analyzed expression of allelic variants of the *Ptprc* gene encoding the membrane phosphatase CD45, expression of which defines all lympho-hematopoietic cells. The *Ptprc^a* allele, encoding CD45.1, and the *Ptprc^b* allele, encoding CD45.2, are easily discriminated by flow cytometry with monoclonal antibodies in congenic mice. Heterozy-gous B6-*Ptprc^{a/b}* mice are expected to express both alleles on all B or T cells, but we were able to detect clearly defined, albeit very rare (~0.01%) subpopulations of B or T cells expressing only one allele or the other (**Figure 7C**; **Figure 7—figure supplement 2A, B**). The monoallelic cells exhib-ited similar staining intensity as homozygous B6-*Ptprc^{a/a}* and B6-*Ptprbc^{b/b}* cells analyzed in parallel. Given the very low frequency of monoallelic *Ptprc* expression, cells lacking CD45 altogether would be predicted to be extremely rare and indeed were not detected. Sorted CD45.1 and CD45.2 single positive T cells from B6-*Ptprc^{a/b}* mice retained monoallelic expression over five- to eightfold expansion after stimulation with anti-CD3/CD28 beads in vitro, demonstrating that RME of *Ptprc* is mitotically stable (**Figure 7D**). Sanger sequencing of reverse-transcribed and amplified RNA isolated from the expanded cells displayed in **Figure 7D** revealed that the monoallelic populations expressed only the allele detected by cell surface staining, demonstrating that the rare observed RME of *Ptprc* reflected transcriptional differences (**Figure 7—figure supplement 2C**). These data argued strongly against the possibility that the monoallelic cells arose due to somatic mutations in one or the other *Ptprc* allele, since most such mutations would not be predicted to disrupt transcription.

Expression of *Cd8a* defines the cytotoxic lineage of T cells. Similar analysis of *Cd8a* for RME employed allele-specific CD8α antibodies. For *Cd8a*, we gated on cells expressing CD8β, the partner chain of CD8α. In (B6 x CBA)F₁ mice, approximately 0.1% of CD8β+ cells lacked one or the other of the CD8α alleles, CD8.1 or CD8.2 (**Figure 7E**; **Figure 7—figure supplement 2D, E**). Sorted single positive cells that retained expression of CD8β after stimulation and expansion retained expression of the initially selected allele of CD8α, demonstrating mitotically stable RME (**Figure 7F**). Finally, analysis of *Thy1*, a marker of T cells in mice, also employed allele-specific antibodies to discriminate the allelic Thy-1.1 and Thy-1.2 proteins, expressed by AKR and B6 strains, respectively. In (B6 x AKR)F₁ hybrids ~5% of CD4+ T cells expressed only one allele or the other (**Figure 7G**; **Figure 7—figure supple-ment 2F, G**). Again, the monoallelic populations displayed an impressive degree of mitotic stability (**Figure 7H**). Hence, *Thy1* represents an RME gene with a remarkably high allelic failure rate. Notably, the *Thy1* promoter contains a CpG island, indicating that the RME we identifiy in lineage-defining genes is not restricted to promoters with low CpG content. In conclusion, RME was detectable for all four genes we examined, all of which were previously considered to be expressed in all cells of the lineages analyzed.

These findings supported the notion that RME is characteristic of many genes and is a natural consequence of enhancer-promoter interactions, rather than a specialized form of gene expres-sion. We quantified allelic failure rates of the various genes examined in this study (**Figure 7—figure supplement 2H**). We propose that in the absence of selection for biallelic expression, most genes

exist along a continuum of allelic failure rates, and that RME and 'non-RME' genes differ quantitatively with respect to allelic failure rates rather than qualitatively with respect to dedicated, RME-specific regulatory programs.

## Discussion

The $Klra1_{Hss1}$ element was previously reported to be a 'switch' element active only in immature, Ly49-negative NK cells (*Saleh et al., 2004*). Our extensive analysis herein demonstrated that Hss1 displays properties of enhancers in mature cells, consistent with the conclusions of others based on reporter analysis (*Gays et al., 2015*). Furthermore, the loss of Ly49G2 expression after deletion of $Klra7_{Hss1}$ that we have documented in mature Ly49G2+ cells is inconsistent with a solely developmental role of Hss1, and further supports its enhancer identity in mature NK cells. Finally, our results showed that variegation arises and is modulated by enhancer deletion (including $Klra7_{Hss5}$ and $Klrk1_{5'E}$) rather than introduction of variegating switch elements.

The main significance of our results is to link RME with the binary model of enhancer action, placing previous observations of enhancer deletion-associated variegation in the context of the pervasive and naturally-occurring RME phenomenon. Remarkably, deletion of an enhancer upstream of the $Klrk1$ gene imparted an RME expression pattern that fully recapitulated the stochasticity, mitotic stability and promoter accessibility features of naturally variegated NK receptor genes. In this instance, enhancer-like elements downstream of $Klrk1$ may suffice to impart the lower frequency of expression. The striking commonalities of enhancer deletion variegation and natural variegation of NK receptor genes and other RME genes argues that RME is an extreme manifestation of the inherent probablistic nature of stable gene activation rather than a specialized mechanism to impose a variegated expression pattern.

The data also reveal the quantitative impact of enhancer strength on allelic expression frequencies. Deletion of $Klra7_{Hss5}$, a relatively minor and constitutively accessible enhancer, reduced the frequency of expression of $Klra7$, a natural RME gene, directly tying the enhancer deletion-associated variegation phenomenon to RME. This result powerfully argues that enhancers are not simply permissive for expression at RME genes, but are also instructive regarding expression probability. We hypothesize that the broad range of frequencies with which different $Klra$ genes are naturally expressed (~5–60%) in large part reflects differences in enhancer strength. Probabilistic enhancer action has also been documented in *Drosophila*, where regulation of the gap genes by multiple enhancers has been suggested to ensure a high probability of gene expression, and removal of one of multiple enhancers was shown to increase gene 'failure rate' (*Perry et al., 2011*). We propose that the binary decision to express a gene is regulated by quantitatively varying enhancer activity, which is comprised of both the number of enhancers acting upon a gene and the strength of individual enhancers within that set. Our results are consistent with recent findings that enhancers are probabilistic regulators of transcription burst frequency rather than burst size (*Bartman et al., 2016*; *Larsson et al., 2019*). How enhancer control of the probability of stable gene expression interfaces with the control of transcription burst frequency is an exciting area for future investigation.

Our data showing RME of the lineage-defining $Klrk1$, $Ptprc$, $Cd8a$, and $Thy1$ genes support the generality of probabilistic gene expression, and suggest that RME is even more prevalent than the previous estimates (*Eckersley-Maslin et al., 2014*; *Gendrel et al., 2014*; *Nag et al., 2013*; *Reinius et al., 2016*). Although RME has been associated previously with poorly expressed genes (*Gendrel et al., 2014*; *Reinius et al., 2016*), our results extend the phenomenon to relatively highly expressed genes. We propose that genes lie along a spectrum of allelic failure rates that are largely controlled by enhancer strength, with documented RME genes on the highest end of that spectrum. RME applies to genes encoding cell surface receptors ($Klra$, $Klrc1$, $Klrk1$ etc.) and to a gene encoding a transcription factor, Bc11b (*Ng et al., 2018*). It applies as well to CpG poor promoters (the aforementioned receptor genes) and to genes that harbor promoter proximal CpG islands ($Thy1$ and the gene encoding Bcl11b *Ng et al., 2018*). High-resolution genome-wide approaches will eventually provide a comprehensive picture of the full extent of RME.

We predict that the mechanism of mitotic stability of active and silent RME alleles is likely related to maintenance of gene expression states broadly, and may involve bookmarking of promoters (*Xu et al., 2017*), rather than repressive chromatin at silent alleles. Indeed, we found that expressed and silent $Klra7$ alleles are distinguished only by the presence of active marks at the promoters and gene bodies

of active alleles. Enhancers are constitutively accessible and activated on expressed and silent alleles alike, consistent with the decoupling of enhancer and promoter accessibility seen at RME loci generally. The absence of a repressive chromatin state on silent NK receptor alleles suggests that the RME 'off state' can reflect a stably inactive, as opposed to repressed, chromatin state. Numerous lineage non-specific genes that are also silent in NK cells (e.g. *Cd19*, *Cd3e*) also lacked traditional repressive chromatin modifications, yet are generally not subject to subsequent activation after an initial failure to be activated. In the case of the NK receptor genes and RME genes broadly it appears that the inactive state is maintained in mature cells in spite of continued enhancer activation, suggesting silent promoters are no longer competent for activation—perhaps due to lack of critical promoter-activating pioneer factor activity or reduced nucleosome remodeling or 'promoter opening' capacity that is only present at sufficient levels for developmentally regulated gene activation during differentiation. The relevant biochemical activity is likely a property of general rather than gene-specific factors, given the apparent pervasiveness of RME even among lineage-defining genes.

The results, in concert with previous studies, provide strong evidence that the principles underlying enhancer deletion-associated variegation and RME are broadly applicable to the regulation of many if not all genes. A possible mechanistic explanation of the probabilistic nature of gene expression in RME is that the initially inactive promoters present an energetic threshold that must be overcome to stably activate gene expression, and that enhancer strength determines the probability of overcoming this threshold. Once overcome, the active state is largely stable even in the absense of the initial signal or enzymatic activity, which may be restricted to a cellular differentiation window. The energy barrier may be due to the initially chromatinized state of the promoter and the energy required for nucleosome remodeling of promoters. It may be this property that is the measure of 'enhancer strength', that is, the capacity of the enhancer to overcome the energy threshold presented by the promoter within a limited window during differentiation. The number of enhancer elements in a gene, the availability of relevant transcription factors that bind the enhancers and the affinity with which the enhancer binds the factors are all likely relevant in determining enhancer strength. It remains possible that in some cases of RME, additional modifications of the promoter may contribute to the energy barrier as well.

Differential DNA methylation of alleles is not broadly associated with all or even most RME genes (*da Rocha and Gendrel, 2019*; *Eckersley-Maslin et al., 2014*), nor does it appear to be responsible for maintenance of silent alleles in the majority of tested RME genes (*Eckersley-Maslin and Spector, 2014*; *Gupta et al., 2022*; *Marion-Poll et al., 2021*). Silent *Klra* and *Klrc1* alleles in mouse NK cells were reported to be hypermethylated relative to active alleles (*Rouhi et al., 2007*; *Rouhi et al., 2006*; *Rouhi et al., 2009*), and our data show that silent *Klrk1* alleles in 5'EΔ/5'EΔ NK cells are also hypermethylated. It is doubtful that DNA methylation is sufficient to maintain the silent state, however, since inhibitors of DNA methyltransferases failed to derepress inactive *Klra* or *Klrc1* genes in cell lines (*Rouhi et al., 2007*; *Rouhi et al., 2006*; *Rouhi et al., 2009*), or other silent RME genes that were hypermethylated (*Eckersley-Maslin and Spector, 2014*; *Gupta et al., 2022*; *Marion-Poll et al., 2021*). In light of the collective data on DNA methylation and silent alleles in RME we suspect it is not generally causative in maintaining RME. Furthermore, the *Klra*, *Klrc1*, and *Klrk1* genes are all hypermethylated on both alleles in progenitor hematopoietic stem cells, suggesting that demethylation is associated with gene activation as opposed to a model where alleles are randomly silenced by de novo DNA methylation. The possibility remains that for these genes, preexisting DNA methylation contributes to the energy barrier that must be overcome by enhancer action, and that demethylation accompanies de novo gene activation. DNA methylation is not a requisite feature of RME promoters, however, as many RME genes are hypomethylated on both alleles.

The RME of the NK receptor genes resembles the monoallelic expression pattern of cytokine genes including *Il2, Il4, Il5, Il10,* and *Il13* (*Bix and Locksley, 1998*; *Gendrel et al., 2016*; *Kelly and Locksley, 2000*; *Rivière et al., 1998*). The cytokine genes are inducible in response to TCR stimulation and therefore expression is inherently unstable, but impressive stability over several mitotic divisions was observed for *Il4* (*Bix and Locksley, 1998*), and the probability of *Il4* allelic activation and biallelic expression correlated with the strength of the inducing signal (*Rivière et al., 1998*). Intriguingly, the *Il4* and *Il13* genes are closely linked and are co-regulated by an enhancer, *CNS1*, that was found to be constitutively acetylated at histone H3, and thus permissive for expression (*Guo et al., 2005*). Expression of the *Il4* and *Il13* gene alleles was independent, however, in a manner strikingly similar to the NK receptor genes. It is probable that the general principles uncovered by us and others apply

to genes that are induced via stimulation as well as genes whose expression is acquired during differentiation. Importantly, enhancer deletions have been observed to reduce the stochastic expression probabilities of the monoallelically expressed odorant receptor genes (*Khan et al., 2011*) and trace amine-associated receptors of the olfactory epithelium (*Fei et al., 2021*). It therefore seems likely that the probabilistic nature of gene activation and binary model of enhancer-promotor communication accounts for the initial stochastic step across many or even all types, even if they might differ with respect to feedback and allelic exclusion mechanisms.

Apparently, RME often occurs at such a low rate that it is both beneath ready detection and presumably irrelevant for the function of a cell lineage. In the case of NK receptors, we propose that evolution has exploited RME to generate a complex combinatorial repertoire of NK cell specificities. More speculatively, by regulating expression of fate-determining mediators, RME may underlie stochastic cell fate decisions in some instances of cellular development (*Ng et al., 2018*). From an evolutionary perspective, appreciable RME of a gene could arise by mutation of strong enhancers of a precursor gene, by providing a new gene with a weak enhancer, or by diminishing the concentration of relevant enhancer-binding transcription factors in a given lineage of cells (as has been shown for several relevant TFs for the *Klra* genes) (*Ioannidis et al., 2003*; *Ohno et al., 2008*).

Finally, our results suggest that allelic failure rates may in some cases dwarf the rates of null alleles generated by somatic mutation. As a novel mechanism of genetic haploinsufficiency at the cellular level, RME might have broad implications in genetic disease etiology and penetrance of disease phenotypes in heterozygous individuals.

## Materials and methods
### Animals and animal procedures
All mice were maintained at the University of California, Berkeley. *Klrk1*$^{-/-}$ mice are available at the Jackson Laboratory (JAX Stock No. 022733). C57BL/6 J (B6), BALBc/J and B6;129-*Ncr1*$^{tm1Oman}$/J (Ncr$^{gfp}$) were purchased from the Jackson Laboratory and bred at UC Berkeley. BALB/cJ, CBA/J and AKR/J and B6.SJL-*Ptprc*$^a$ *Pepc*$^b$/BoyJ mice were purchased from the Jackson Laboratory. F$_1$ hybrid mice were purchased from the Jackson Laboratory or were generated at UC Berkeley from inbred parents.

For the generation of CRISPR edited mice, Cas9 RNP was delivered to single-cell embryos either through microinjection or CRISPR-EZ electroporation, both of which are described in reference (*Modzelewski et al., 2018*). *Klra1*$_{Hss1\Delta}$ mice were generated by microinjection, while *Klrc1*$_{5'E\Delta}$, *Klrk1*$_{5'E\Delta}$ and *Klra7*$_{Hss1\Delta}$ mice were generated by CRISPR-EZ electroporation. Whether through microinjection or electroporation, we used paired sgRNAs flanking the enhancer to generate enhancer deletion mice. sgRNAs were selected using the GPP web portal from the Broad Institute. Guides with highest predicted editing efficiencies were prioritized, while also minimizing for predicted off-target cutting in protein-coding genes. sgRNAs were generated using the HiScribe T7 Quick High Yield RNA Synthesis Kit (New England Biolabs). Founder mice (F$_0$) harboring deletion alleles were backcrossed to C57BL/6 J (B6) mice to generate heterozygous F$_1$ mice, and were then intercrossed to generate WT, heterozygous and homozygous littermates for experiments. All sgRNAs used for the generation of enhancer deletion mice are listed in *Supplementary file 1*. Primers used to PCR identify edited founders and genotype subsequent filial generations are listed in *Supplementary file 1*. All animals were used between 8 and 32 weeks of age, and all experiments were approved by the UC Berkeley Animal Care and Use Committee (ACUC).

### Flow cytometry
Single-cell splenocyte suspensions were generated by passing spleens through a 40 µm filter. Red blood cells were lysed with ACK buffer. Fresh splenocytes, or where indicated cells cultured with 1000 U/ml recombinant human IL-2 (National Cancer Institute) were stained for flow cytometry in PBS containing 2.5% FCS (FACS Buffer). Before staining with antibodies, FcγRII/III receptors were blocked for 15 min at 4°C using 2.4G2 hybridoma supernatant. Cells were washed with FACS buffer and then stained with antibodies directly conjugated to fluorochromes or biotin at 4°C for 15–30 min. In order to differentiate between alleles of a receptor in (B6 x BALB/c)F$_1$ hybrid NK cells, the B6-specfic clone was used first in order to block epitopes in competition with the clone recognizing both alleles. For example, to discriminate Ly49G2 alleles, cells were stained for at least 15 min with 3/25 which

recognizes Ly49G2[B6], and then 4D11 (which recognizes both alleles) was added. For discriminating alleles of NKG2A, cells were stained first with the NKG2A[B6]-specific 16a11, followed by 20d5, which binds to both alleles. Anti-Ly49A[B6] (A1) was added before the non-discriminating JR9 mAb, but in this case, cells expressing only the B6 allele did not resolve from the population of cells expressing both alleles. When necessary, cells were washed and then stained with secondary antibody or fluorochrome-conjugated streptavidin. Near-IR viability dye (Invitrogen L34975) or DAPI (Biolegend 422801) were used to discriminate live cells. Flow cytometry was carried out using an LSR Fortessa or X20 from BD Biosciences, and data were analyzed using FlowJo software. In all cases, NK cells were defined as CD3-NKp46+ splenocytes. For sorting on a BD FACSAria II sorter, the samples were prepared nearly identically as they were for flow cytometric analysis with the exception that the medium used was sterile RPMI 1640 (ThermoFisher) with 5% FCS.

## Antibodies used in flow cytometry
From Biolegend: anti-CD3ε (145–2C11) PE-Cy5, anti-CD19 (6D5) PE-Cy5, anti-F4/80 (BM8) PE-Cy5, anti-Ter119 (TER-119) PE-Cy5, anti-NKp46 (29A1.4) BV421, anti-NKG2A[B6] (16a11) PE, anti-Ly49A[B6] (A1) PE, anti-NKG2D (CX5) PE/Dazzle 594, anti-CD8β (YTS156.7.7) PE-Cy7, anti-CD45.1 (A20) APC, anti-CD45.2 (104) FITC, anti-CD90.2 (53–2.1) PE or FITC, goat-anti-mouse IgG (Poly4053) PE. From eBioscience/ThermoFisher: anti-NKG2A (20d5) PerCP, anti-Ly49I (YLI-90) FITC, anti-Ly49G2 (4D11) PerCP-eFlour 710 or PE-Cy7, anti-CD90.1 (HIS51) FITC, anti-rat IgG F(ab')2 (polyclonal, lot 17-4822-82) APC. From BD Biosciences: anti-CD4 (GK1.5) BUV737. From BioXCell: anti-CD8.1 (116–13.1) unconjugated, anti-CD8.2 (2.43) unconjugated. Prepared in-house: anti-Ly49A (JR9) (*Roland and Cazenave, 1992*) biotin, anti-Ly49G2[B6] (3/25) (*Tanamachi et al., 2001*), anti-NKG2D (MI-6) biotin. Biotin conjugated antibodies were used in conjunction with streptavidin (conjugated with APC or PE) from Biolegend.

## Ex vivo NK cell cultures
NK cells were prepared from spleens by passage through a 40 µm filter. Red blood cells were lysed with ACK. Splenocytes were cultured in RPMI 1640 (ThermoFisher) with 1000 U/mL IL-2 (National Cancer Institute) and 5% FCS. In all cases, media was supplemented with 0.2 mg/mL glutamine (Sigma), 100 U/mL penicillin (ThermoFisher), 100 µg/mL streptomycin (Thermo Fisher Scientific), 10 µg/mL gentamycin sulfate (Fisher Scientific), 50 µM β-mercaptoethanol (EMD Biosciences), and 20 mM HEPES (ThermoFisher).

## Analysis of the stability of monoallelic expression of NKG2D
NKG2D+/- NK cells were sorted from WT or *Klrk1^{5'EΔ/5'EΔ}* mice on day 2 or 3 of ex vivo NK cell culture in IL-2 medium as described above. Cells were cultured in vitro in IL-2 containing media for a further 8–10 days, during which cells expanded ~10–100 fold based on hemocytometer counts. Cells were analyzed for NKG2D expression by flow cytometry. In all cases medium contained 5% FCS (Omega Scientific), 0.2 mg/mL glutamine (Sigma), 100 U/mL penicillin (ThermoFisher), 100 µg/mL streptomycin (Thermo Fisher Scientific), 10 µg/mL gentamycin sulfate (Fisher Scientific), 50 µM β-mercaptoethanol (EMD Biosciences), and 20 mM HEPES (ThermoFisher). Cells were incubated at 37°C in 5% $CO_2$.

## Ex vivo assay for the stability of monoallelic expression in T cells
Cells from the spleens and a collection of lymph nodes (brachial, axial, inguinal, mesenteric) from $F_1$ hybrid mice and parental inbred line controls were combined and passed through a 40 µm filter, and red blood cells were lysed with ACK buffer. Cells were prepared for sorting as described above, staining with the relevant allele-specific antibodies. For CD45 monoallelic expression, Thy1+ cells were further gated according to CD45 allelic expression. For CD8α monoallelic expression, CD3+CD8β+ cells were analyzed for CD8α allelic expression. For Thy1 monoallelic expression, CD4+MHC II- cells were analyzed for Thy-1 allelic expression. Cells expressing either the paternal or maternal allele (or both) of the receptor studied were sorted and expanded for 1 week in RPMI 1640 (ThermoFisher) containing 200 U/mL recombinant IL-2, Dynabeads mouse T-activator CD3/CD28 (ThermoFisher) beads at a 1:1 cells to beads ratio, 10% FCS, and supplemented as described for ex vivo NK cell cultures. After 1 week of expansion, cells were harvested, counted by hemocytometer and prepared for a second sort. After sorting for expression of the relevant receptor allele again in order to ensure purity, cells were

once again expanded in a restimulation, this time with a cells to beads ratio of 10:1. After the second expansion, cells were again counted, stained and prepped for final analysis of monoallelic receptor expression by flow cytometry.

In analysis of *Ptprc* monoallelic expression, RNA was isolated from expanded T cells expression either or both *Ptprc* alleles as displayed in *Figure 7D* using the iScript cDNA synthesis kit (BioRad), from 10,000 to 40,000 cells. Half of the reaction volume (10 μL out of 20 μL) was used to PCR amplify a region of the *Ptprc* transcript using intron-spanning PCR primers (*Supplementary file 1*).

### Enhancer deletion in primary NK cells via Cas9-RNP nucleofection

Ex vivo editing of primary mouse NK cells was carried out according to a modified version of the protocol used to modify primary human T cells described in reference (*Roth et al., 2018*). Cas9 was purchased from the UC Berkeley Macro Lab core (40 μM Cas9 in 20 mM HEPES-KOH, pH 7.5, 150 mM KCl, 10% glycerol, 1 mM DTT), and sgRNAs were transcribed in vitro according to the Corn lab online protocol (https://www.protocols.io/view/in-vitro-transcription-of-guide-rnas-and-5-triphos-bqjbmuin). NK cells were prepared by sorting day 5 IL-2 cultured NK cells from (B6 x BALB/c)F$_1$ hybrids. CD3-NKp46+ cells were sorted to be positive for either NKG2A$^{B6}$ using the 16a11 clone or Ly49G2$^{B6}$ using the 3/25 clone, and cells were further cultured overnight in RPMI 1640 media containing 5% FCS and 1000 U/mL IL-2 (National Cancer Institute). On day 6, 1 million sorted NK cells were prepared for nucleofection using the Lonza 4D-Nucleofector per condition. Cas9 and sgRNAs were complexed at a molar ratio of 1:2 (2.5 μL of 40 μM Cas9 was added to 2.5 μL of sgRNA suspended at 80 μM (6.5 μg) in nuclease-free H$_2$O). If two flanking guides were used, 1.25 μL of each were used, maintaining the Cas9 to sgRNA molar ratio. Cas9-RNP was complexed for 15 min at 37°C and transferred to a single well of a 96-well strip nucleofection cuvette from Lonza for use with the Nucleofector 4D. 1 million sorted day 6 IL-2 cultured NK cells were resuspended in 18 μL of supplemented Lonza P3 buffer from the P3 Primary Cell kit, and added to the Cas9-RNP complex. Cells were nucleofected using the CM137 nucleofection protocol and 80 μL pre-warmed RPMI 1640 with 5% FCS was immediately added. After a 15-min recovery period at 37°C, cells were returned to culture in 1 mL of RPMI 1640 with 5% FCS and 1000 U/mL IL-2. After 5–7 days in culture maintaining a maximum density of 1 million cells/mL, receptor expression was assayed by flow cytometry. In order to validate enhancer flanking guides (*Supplementary file 1*) an identical protocol was followed with either day 5 IL-2 cultured splenocytes, or day 5 IL-2 cultured NK cells isolated using the MojoSort NK isolation kit from Biolegend, but instead of analysis by flow cytometry, gDNA was prepared and used as a template for PCR to detect the expected deletion.

### F$_1$ hybrid genetics and calculations of expected changes in receptor-expressing NK cell populations

F$_1$ hybrid genetics were carried out by breeding WT or CRISPR/Cas9-edited males on the B6 background to females from the following backgrounds: BALBc/J, CBA/J, AKR/J. Edited alleles were crossed only to BALBc/J, while CBA/J and AKR/J were used in the F$_1$ hybrid analysis of monoallelic expression of CD8α and Thy-1, respectively.

We estimated the expected frequencies of NK cells in (*Klrc1$^{B6-5'EΔ/BALB/c+}$*) F$_1$ mice by assuming independence of allelic expression. That assumption leads to the following predictions:

The percentage of cells expressing neither allele in the mutant will equal the sum of the percentages of the two NK cell populations that lack NKG2A$^{BALB/c}$ in WT (B6 x BALB/c)F$_1$ hybrids, that is the cells that express neither allele, and cells expressing only the B6 allele.

The percentage of cells expressing NKG2A$^{BALB/c}$ only in the mutant will equal the sum of the percentages of the NK cell populations in WT (B6 x BALB/c)F$_1$ hybrids that express NKG2A$^{BALB/c}$, that is the cells that express only the BALB/c allele, and the cells expressing both alleles.

The percentages of cells expressing NKG2A$^{B6}$ only or both NKG2A$^{B6}$ and NKG2A$^{BALB/c}$ will be 0, since NKG2A$^{B6}$ is not expressed.

The expected frequency of cells expressing NKG2D in *Klrk1$^{-/5'EΔ}$* mice was calculated assuming stochastic expression of alleles, and was based on the frequency of cells expressing NKG2D, or not, in *Klrk1$^{5'EΔ/5'EΔ}$* mice. The frequency of cells lacking expression of a given allele is the square root of the frequency of cells expressing neither allele. Subtraction of this proportion from 1 yields the predicted

frequency of cells expressing NKG2D in $Klrk1^{-/5'E\Delta}$ mice. For example, an observed NKG2D expression frequency of ~67% in a $Klrk1^{5'E\Delta/5'E\Delta}$ mouse would result in an expected frequency datapoint of ~43%.

The expected changes in populations with respect to Ly49G2 alleles in $Klra7^{B6-Hss5\Delta/BALB/c+}$ mice were calculated with the same assumption of independent regulation of alleles.

We started by calculating the overall percentage of cells expressing Ly49G2$^{B6}$ in the $F_1$ mice with the mutation, which averaged 47.7% of that in WT $F_1$ mice.

The predicted percentage of cells expressing only Ly49A$^{B6}$ in the mutant $F_1$ was then 47.7% of the percentage of cells expressing only Ly49A$^{B6}$ in WT mice.

And the predicted percentage of cells expressing both alleles in the mutant $F_1$ was 47.7% of the percentage of cells expressing both alleles in WT mice.

The predicted percentage of cells expressing neither allele in the mutant $F_1$ was calculated as the percentage of cells expressing neither allele in WT mice plus 52.3% (100%–47.7%) of the percentage of NK cells that express only Ly49G2$^{B6}$ in WT mice.

Finally, the predicted percentage of NK cells expressing only Ly49G2$^{BALB/c}$ in the mutant was calculated as the percentage expressing only Ly49G2$^{BALB/c}$ in WT mice plus 52.3% of the NK cells expressing both alleles in WT mice.

Note that the genetic background of the mice significantly influences *Klra7* expression even in WT mice, presumably reflecting trans-acting events (e.g. each WT *Klra7*$^{B6}$ allele is expressed on ~31% of NK cells in B6 mice, but only ~19% in $F_1$ hybrid mice). Therefore expected data are calculated using Ly49G2$^{B6}$ expression frequencies in $Klra7^{B6-5'E+/BALB/c+}$ mice.

## ATAC-Seq

ATAC-seq was performed as previously described in reference (*Buenrostro et al., 2013*). Briefly, 50,000 sorted NK cells were washed in cold PBS and resuspended in lysis buffer (10 mM Tris-HCl, pH 7.4; 10 mM NaCl; 3 mM MgCl$_2$; 0.1% (v/v) Igepal CA-630). The crude nuclear prep was then centrifuged and resuspended in 1 x TD buffer containing the Tn5 transposase (Illumina FC-121–1030). The transposition reaction was incubated at 37°C for 30 min and immediately purified using the Qiagen MinElute kit. Libraries were PCR amplified using the Nextera complementary primers listed in reference (*Buenrostro et al., 2013*) and were sequenced using an Illumina NextSeq 500 or a HiSeq 4000.

## CUT&RUN

CUT&RUN was performed essentially as previously described (*Skene et al., 2018*). Briefly, 50,000–500,000 NK cells sorted as indicated were washed and immobilized on Con A beads (Bangs Laboratories) and permeabilized with wash buffer containing 0.05% w/v Digitonin (Sigma-Aldrich). Cells were incubated rotating for 2 hr at 4°C with antibody at a concentration of 10–20 µg/mL. Permeabilized cells were washed and incubated rotating at room temperature for 10 min with pA-MNase (kindly provided by the Henikoff lab) at a concentration of 700 ng/mL. After washing, cells were incubated at 0°C and MNase digestion was initiated by addition of CaCl$_2$ to 1.3 mM. After 30 min, the reaction was stopped by the addition of EDTA and EGTA. Chromatin fragments were released by incubation at 37°C for 10 min, purified by overnight proteinase K digestion at a concentration of 120 µg/mL with 0.1% wt/vol SDS at 55°C. DNA was finally purified by phenol/chloroform extraction followed by PEG-8000 precipitation (final concentration of 15% wt/vol) using Sera-mag SpeedBeads (Fisher) (https://ethanomics.files.wordpress.com/2012/08/serapure_v2-2.pdf).

Libraries were prepared using the New England Biolabs Ultra II DNA library prep kit for Illumina as described online (https://www.protocols.io/view/library-prep-for-cut-amp-run-with-nebnext-ultra-ii-bagaibse?version_warning=no) with the following specifications and modifications. The entire preparation of purified CUT&RUN fragments from a reaction were used to create libraries. For histone modifications, end repair and dA-tailing were carried out at 65°C. NEB hairpin adapters (From NEBNext Multiplex Oligos for Illumina) were diluted 25-fold in TBS buffer and ligated at 20°C for 15 min, and hairpins were cleaved by the addition of USER enzyme. Size selection was performed with AmpureXP beads (Agencourt), adding 0.4 X volumes to remove large fragments. The supernatant was recovered, and a further 0.6 X volumes of AmpureXP beads were added along with 0.6 X volumes of PEG-8000 (20% wt/vol PEG-8000, 2.5 M NaCl) for quantitative recovery of smaller fragments. Adapter-ligated libraries were amplified for 15 cycles using NEBNext Ultra II Q5 Master Mix using the universal primer and an indexing primer provided with the NEBNext oligos. Amplified libraries were further purified

with the addition of 1.0 X volumes of AmpureXP beads to remove adapter dimer and eluted in 25 µL $H_2O$. Libraries were quantified by Qubit (ThermoFisher) and Bioanalyzer (Agilent) before sequencing on an Illumina HiSeq 4,000 or MiniSeq as paired-ends to a depth of 10–32 million.

The following antibodies were used for CUT&RUN: Abcam: anti-H3K4me1(ab8895), anti-H3K4me2 (ab7766), anti-H3K4me3 (ab8580), anti-H3K27ac (ab4729), anti-H3K9me3 (ab8898). Cell Signaling: anti-H3K27me3 (C36B11), anti-H2AUb1 (D27C4). Control IgG (cIgG) from Biolegend: Mouse IgG2aκ (MOPC-173).

## Bisufite conversion and analysis of promoter CpG methylation

Genomic DNA was extracted from 50,000 to 100,000 FACS-sorted cells by addition of 10 µl of water, 10 µl PBS, 5 µl proteinase K, and 15 µl lysis buffer. The lysate was incubated for 30 min at 58°C. Bisulfite conversion was performed using the EpiTect Fast LyseAll Bisulfite Kit (Qiagen) according to the manufacturer's protocol. Cleanup was also performed according to the manufacture's protocol using MinElute columns. Bisulfite-treated DNA was amplified by PCR using EpiTaq HS (Takara) using the primers targeting a region spanning 481bp and 5 CpG sites in the the *Klrk1* promoter indicated in *Figure 5—figure supplement 2A*: (Forward: 5'- ATAGAGATAGTAGAAAAAAAATTTGTTAGAAT – 3'; Reverse: 5'-AAAACTTTCCACAATCTCAAAAACTAAATT - 3'). 35 amplification cycles were performed: 10s at 98°C, 30s at 55°C, and 60s at 72°C. PCR products were cloned into the TOPO TA plasmid from Invitrogen, and transformed into XL-1 competent cells (UC Berkeley Macro Lab). Clones were selected by blue/white screening, used for colony PCR, and analyzed by Sanger sequencing of the colony PCR product.

We further analyzed the CpG methylation status of the *Klrk1* promoter in hematopoietic stem cells (HSCs) by mining published whole genome bisulfite sequencing (WGBS) data from *Li et al., 2021*. We downloaded .cov files from the associated GEO series (GSE167237), and scored the frequency of methylated CpGs at the same five sites that we analyzed in the bisulfite analysis described above. This series contains 4 WGBS datasets generated using E14 fetal liver HSCs and 2 generated using adult bone marrow HSCs; datapoints were compiled across all six datasets. We performed a similar analysis at the *Klra1* and *Klra7* promoters in a region spanning 500 bp around each promoter.

## Datasets and processing and visualization

Raw mined datasets were downloaded from NCBI Gene Expression Omnibus (GEO) or the European Bioinformatics Institute (EBI). NK cell ATAC-seq and histone modification (H3K4me1, H3K4me2, H3K4me3, H3K27ac) were from reference (*Lara-Astiaso et al., 2014*) under GEO accession numbers GSE59992 and GSE60103. Runx3 ChIP-seq data and non-immune serum control in NK cells were sourced from reference (*Levanon et al., 2014*) (GSE52625) and T-bet ChIP-seq data and input control were sourced from reference (*Shih et al., 2016*) (GSE77695). p300 ChIP-seq raw data was sourced from reference (*Sciumè et al., 2020*) (GSE145299). p300 ChIP-seq peaks were called in reference (*Sciumè et al., 2020*) and downloaded in .csv format.

Raw data from all datasets (mined or generated in this study) were processed using a pipeline assembled in-house. Datasets were tested with FastQC. Paired-end reads were then aligned to the mm10 reference genome using Bowtie2 with the `--sensitive` parameter. Paired-end CUT&RUN libraries were tested and aligned with the same pipeline. All reads aligned to the mitochondrial chromosome were removed with samtools. Aligned reads were then sorted, indexed, and filtered for a mapping quality of ≥10 with samtools. PCR duplicates were removed with Picard (Broad Institute). Reads covering blacklisted regions (ENCODE mm10 database) were removed with bedtools. Data were then normalized to signal per million reads (SPMR) when calling narrow peaks with MACS2. Resultant bedgraph files were converted to bigwigs with the bedGraphToBigWig program from the UCSC Genome Browser toolkit for visualization on Integrative Genomics Viewer (IGV) (*Thorvaldsdóttir et al., 2013*). Data in *Figure 1—figure supplement 1A* were plotted using the Bioconductor package SeqPlots (*Stempor and Ahringer, 2016*).

## Ranking of accessible sites in NK cells according to H3K4me1:me3 ratio

Reads from duplicate ChIP-seq datasets (for both H3K4me1 and H3K4me3) from reference (*Lara-Astiaso et al., 2014*) were merged to ensure robust signal, and the resultant files were processed and normalized as above. NK cell ATAC-seq peaks were called in the Ly49G2[B6+BALB+] NK cell ATAC-seq

dataset using macs2 narrowpeaks. Before ranking, ATAC-seq peaks were filtered such that only peaks that fell within the top 95% of both H3K4me1 and H3K4me3 signal computed over a 2 kb window from the peak midpoint using pandas and numpy in Python 3.7.4, resulting in 51,650 usable peaks. H3K4me1:me3 raw ratio and log2 ratio bigwigs were generated with the bamCompare utility from deepTools (v2.5.4). The log2 ratio track was visualized on IGV, and the raw ratio was used to rank ATAC-seq peaks. Heatmaps were generated with the computeMatrix and plotHeatmap utilities from deepTools (v2.5.4). Heatmaps were sorted by the mean H3K4me1:me3 ratio signal over a 2 kb window centered at the midpoint of the 51,650 ATAC-seq peaks. Hss1 and 5′E enhancer regions and corresponding promoters at NKC genes were individually predefined and the position of each was then marked on the heatmap.

## Definition of NK cell promoters and enhancers and ranking of regulatory elements according to H3K4me1:me3 ratio

Annotated mouse promoters (defined as the TSS at a single nucleotide) in the mm10 genome assembly were downloaded as a BED file from the EDPNew database (*Dreos et al., 2017*). To identify likely active promoters in NK cells, broad regions of H3K27ac were called based on ChIP-seq data sourced from reference (*Lara-Astiaso et al., 2014*) using the "macs2 callpeak --broad" command. Mouse EDPNew promoters falling within broad H3K27ac domains were identified using the "bedtools intersect -wa" command, resulting in a set of 9901 active promoters in mouse NK cells (*Source data 1*-mouse NK cell promoters).

Enhancers in naïve mouse NK cells were defined as the intersection of ATAC-seq and p300 peaks not found at the promoters as defined above. p300 ChIP-seq peaks in resting NK cells were previously defined and downloaded from reference (*Sciumè et al., 2020*). ATAC-seq peaks that were enriched in p300 binding were identified using the "bedtools intersect -wa" command. To define enhancers that do not overlap annotated promoters, EDPNew promoters were subtracted from p300-enriched ATAC-seq peaks using the "bedtools subtract" command resulting in 10,246 NK cell enhancers (*Source data 1*-mouse NK cell enhancers).

## SNPsplit chromosome of origin reads analysis

Delineation of allele-informative reads was performed similarly as in reference (*Xu et al., 2017*). SNPs between the C57BL/6 (B6) and BALB/cJ (BALB) mouse strains were sourced from the Wellcome Sanger Institute Mouse Genomes Project dbSNP (v142). In order to perform unbiased alignment of reads originating from both the B6 and BALB genomes, SNPs marked by the database were replaced by 'N' in the mm10 reference genome that we use for alignment using SNPsplit (Babraham Institute) (*Krueger and Andrews, 2016*). ATAC-seq datasets generated in (B6 x BALB) $F_1$ hybrid NK cells were then aligned to the N-masked genome using bowtie2 and further processed and normalized as above. Reads that overlapped the annotated SNPs were marked as allelically informative reads after alignment and quality control using SNPsplit. Allele-informative reads were then processed and normalized as described above. ~4% of ATAC-seq reads across the dataset were allele-informative.

## ChromHMM construction of three-state model

CUT&RUN data for four histone modifications (H3K4me3, H3K9me3, H3K27ac, H2AK119ub1) generated in cells expressing neither allele (DN) or both alleles (DP) of Ly49G2 were separately used to construct chromatin states using ChromHMM (v1.22) (*Ernst and Kellis, 2012*). The genome was segmented into three distinct states: state 1 (active chromatin; enriched in H3K4me3 and H3K27ac), state 2 (inactive chromatin; lacking enrichment of all four marks), and state 3 (repressed chromatin enriched in H3K9me3 and H2AK119ub1). The resultant .bed file outputs were visualized with IGV. Chromatin states for both Ly49G2 DN and DP cells are provided in *Source data 4*.

## Statistical analysis

In vivo germline-edited mouse data were compared with one-way ANOVAs with Tukey's multiple comparisons (when three or more genotypes were compared) or student's *t*-tests (when only two groups were compared). Ex vivo edited NK cell experiments were analyzed by ratio paired *t*-tests comparing experimental and control samples within a single experiment. In all cases, *$p < 0.05$; **$p < 0.01$; ***$p < 0.001$; **** $< 0.0001$.

## Acknowledgements

We thank Lily Zhang and Erik Seidel for assistance, Hector Nola and Alma Valeros in the Cancer Research Laboratory at UC Berkeley for expert assistance with flow cytometry and cell sorting, and M Wong for assistance with some Illumina sequencing experiments. We thank Drs. Kathleen Pestal, Han-Yu Shih and Giuseppe Sciume for discussion and feedback. We thank Drs. Jasper Rine, Russell Vance, Ellen Robey, Michel DuPage, and David Martin for critical evaluation of the manuscript.

## Additional information

### Funding

| Funder | Grant reference number | Author |
|---|---|---|
| National Institute of Allergy and Infectious Diseases | R01-AI113041 | David H Raulet |
| National Institute of Allergy and Infectious Diseases | Intramural Research Program of NIH | Stefan A Muljo |
| National Science Foundation | DGE 1752814 | Natalie K Wolf |
| Cancer Research Coordinating Committee | Predoctoral fellowship | Djem U Kissiov |
| National Institute of Allergy and Infectious Diseases | NIAID | Stefan A Muljo |

The funders had no role in study design, data collection and interpretation, or the decision to submit the work for publication.

### Author contributions

Djem U Kissiov, Conceptualization, Data curation, Formal analysis, Funding acquisition, Investigation, Methodology, Resources, Software, Validation, Visualization, Writing - original draft, Writing - review and editing; Alexander Ethell, Data curation, Formal analysis, Methodology, Validation, Visualization; Sean Chen, Investigation, Methodology, Resources; Natalie K Wolf, Data curation, Investigation, Methodology, Writing - review and editing; Chenyu Zhang, Katrine N Madsen, Investigation, Methodology, Writing - review and editing; Susanna M Dang, Yeara Jo, Ishan Paranjpe, Investigation, Methodology; Angus Y Lee, Methodology, Resources, Writing - review and editing; Bryan Chim, Formal analysis, Investigation, Methodology, Writing - review and editing; Stefan A Muljo, Methodology, Project administration, Resources, Writing - review and editing; David H Raulet, Conceptualization, Data curation, Formal analysis, Funding acquisition, Investigation, Methodology, Project administration, Resources, Software, Supervision, Validation, Visualization, Writing - original draft, Writing - review and editing

### Author ORCIDs

Djem U Kissiov http://orcid.org/0000-0001-6279-342X
David H Raulet http://orcid.org/0000-0002-1257-8649

### Ethics

This study was performed strictly and in full accordance with the guidelines set forth by the University of California, Berkeley's Animal Care and Use Committee (ACUC). The protocol was approved by ACUC under the protocol number # AUP-2015-10-8058-1. All experiments were performed on mice euthanized according to the protocol.

### Decision letter and Author response

Decision letter https://doi.org/10.7554/eLife.74204.sa1
Author response https://doi.org/10.7554/eLife.74204.sa2

## Additional files

### Supplementary files

• Supplementary file 1. Guides and primers used to generate and genotype CRISPR/Cas9-edited mice. Guide RNAs (sgRNAs) used to generate germline enhancer deletion mice via electroporation or microinjection are displayed. A flanking guide pair was used to delete the indicated enhancer, except for in the case of $Klrk1_{5'E}$, where two sets of flanking guides were used (all four sgRNAs were simultaneously delivered to embryos). Primers used to genotype mice carrying a deletion allele and mice lacking a WT allele are also shown. These primers allow delineation of WT, heterozygous and homozygous enhancer deletion animals with respect to the indicated enhancer element. More than one primer is shown if PCR was performed as a nested reaction; "1" indicates use in the first amplification and "2" indicates use in the subsequent amplification.

• Supplementary file 2. Guides and primers used for ex vivo NK cell editing. Guide RNAs (sgRNAs) used in the ex vivo NK cell enhancer deletion assay are displayed. Non-targeting sgRNA pairs 1 (nt1) and 2 (nt2) were used as negative controls in *Figure 3—figure supplement 1A*. Primers used to detect the presence of the intended deletion in nucleofected NK cells using the indicated sgRNAs used are also shown. More than one primer is shown if PCR was performed as a nested reaction; "1" indicates use in the first amplification and "2" indicates use in the second amplification.

• Supplementary file 3. Primers used to amplify *Ptprc* PCR products. Intron-spanning primers detecting a region of the *Ptprc* transcript containing 3 allele-informative SNPs.

• Transparent reporting form

• Source data 1. NK cell promoters and enhancers in Figure 1. xlsx file containing a list of called NK cell promoters (first tab), NK cell enhancers (second tab), 51,650 ATAC-seq peaks (1kb windows centered at ATAC-seq peak midpoints over which the H3K4me1:H3K4me3 ratio was calculated) in NK cells (third tab), and labeled promoters and enhancers investigated in this study (fourth tab). See methods for a description of how these lists were generated.

• Source data 2. Original gel images in Figure 1—figure supplement 2. Original agarose gel images corresponding to Figure 1—figure supplement 2 E and H are provided in both annotated and unannotated format. The relevant lanes are highlighted and labeled. The cropped lanes are denoted with a white vertical bar.

• Source data 3. All raw data values used to generate bar graphs. The raw data provided correspond to all graphed data points in both the main and supplementary figures. Each tab corresponds to one or multiple panels within a figure. In cases where the data displayed in a graph are combined from two independent experiments, data points from each experiment are indicated. The results from statistical tests performed are also displayed.

• Source data 4. Chromatin states in Ly49G2 negative and biallelic sorted NK cells in Figure 6—figure supplement 1. .xlsx file containing ChromHMM-derived 3 state model for cells expressing neither (Ly49G2 DN) or both (Ly49G2 DP) NK cells. Models were built using CUT&RUN data as is described in the methods.

### Data availability

Sequencing data have been deposited in GEO under the accession code GSE181197.

The following dataset was generated:

| Author(s) | Year | Dataset title | Dataset URL | Database and Identifier |
|---|---|---|---|---|
| Muljo SA, Raulet DH | 2021 | Binary outcomes of enhancer activity underlie stable random monoallelic expression | https://www.ncbi.nlm.nih.gov/geo/query/acc.cgi?acc=GSE181197 | NCBI Gene Expression Omnibus, GSE181197 |

The following previously published datasets were used:

| Author(s) | Year | Dataset title | Dataset URL | Database and Identifier |
|---|---|---|---|---|
| Lara-Astiaso D, Weiner A, Lorenzo-Vivas E, Zaretsky I, Adhemar Jaitin D, David E, Keren-Shaul H, Mildner A, Winter D, Jung S, Friedman N, Amit I | 2014 | Chromatin state dynamics during blood formation (ATAC-seq) | https://www.ncbi.nlm.nih.gov/geo/query/acc.cgi?acc=GSE59992 | NCBI Gene Expression Omnibus, GSE59992 |
| Lara-Astiaso D, Weiner A, Lorenzo-Vivas E, Zaretsky I, Adhemar Jaitin D, David E, Keren-Shaul H, Mildner A, Winter D, Jung S, Friedman N, Amit I | 2014 | Chromatin state dynamics during blood formation | https://www.ncbi.nlm.nih.gov/geo/query/acc.cgi?acc=GSE60103 | NCBI Gene Expression Omnibus, GSE60103 |
| Levanon D, Negreanu V, Lotem J, Bone KR, Brenner O, Leshkowitz D, Groner Y | 2014 | Runx3 Regulates Interleukin-15-Dependent Natural Killer Cell Activation | https://www.ncbi.nlm.nih.gov/geo/query/acc.cgi?acc=GSE52625 | NCBI Gene Expression Omnibus, GSE52625 |
| Shih HY, Sciume G, Mikami Y, Guo L, Sun HW, Brooks SR, Urban JF, Davis FP, Kanno Y, O'Shea JJ | 2016 | Developmental Acquisition of Regulomes Underlies Innate Lymphoid Cell Functionality | https://www.ncbi.nlm.nih.gov/geo/query/acc.cgi?acc=GSE77695 | NCBI Gene Expression Omnibus, GSE77695 |
| Shih HY, Sciume G, Mikami Y, Nagashima H, Sun HW, Brooks SR, Jankovic D, Yao C, Villarino A, Davis FP, Kanno Y, O'Shea JJ | 2020 | Rapid remodeling of poised chromatin landscapes and transcription factor repurposing facilitate gene induction in natural killer cells | https://www.ncbi.nlm.nih.gov/geo/query/acc.cgi?acc=GSE145299 | NCBI Gene Expression Omnibus, GSE145299 |
| Zhou J, Liu D | 2021 | DNA methylation regulates haematopoietic development | https://www.ncbi.nlm.nih.gov/geo/query/acc.cgi?acc=GSE167237 | NCBI Gene Expression Omnibus, GSE167237 |

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

# Appendix 1

## Appendix 1—key resources table

| Reagent type (species) or resource | Designation | Source or reference | Identifiers | Additional information |
|---|---|---|---|---|
| Genetic reagent (*Mus musculus*) | C57BL/6J background | Jackson Laboratory | Strain #:000664 RRID:IMSR_JAX:000664 | |
| Genetic reagent (*Mus musculus*) | BALB/cJ background | Jackson Laboratory | Strain #:000651 RRID:IMSR_JAX:000651 | |
| Genetic reagent (*Mus musculus*) | AKR/J | Jackson Laboratory | Strain #:000648 RRID:IMSR_JAX:000648 | |
| Genetic reagent (*Mus musculus*) | CBA/J | Jackson Laboratory | Strain #:000656 RRID:IMSR_JAX:000656 | |
| Genetic reagent (*Mus musculus*) | CB6F1/J | Jackson Laboratory | Strain #:100007 RRID:IMSR_JAX:100007 | Purchased from the Jackson Laboratory or generated in-house |
| Genetic reagent (*Mus musculus*) | B6CBAF1/J | Jackson Laboratory | Strain #:100011 RRID:IMSR_JAX:100011 | |
| Genetic reagent (*Mus musculus*) | B6AKRF1/J | Jackson Laboratory | | Generated in-house by crossing C57BL/6J with AKR/J |
| Genetic reagent (*Mus musculus*) | B6.Cg-$Klrk1^{tm1Dhr}$/J | Jackson Laboratory | Strain #:022733 RRID:IMSR_JAX:022733 | |
| Genetic reagent (*Mus musculus*) | B6-$Klra1_{Hss1Δ}$ | This paper | | Generated by Cas9 RNP microinjection |
| Genetic reagent (*Mus musculus*) | B6-$Klrc1_{5'EΔ}$ | This paper | | Generated by Cas9 RNP electroporation (CRISPR-EZ) |
| Genetic reagent (*Mus musculus*) | B6-$Klrk1_{5'EΔ}$ | This paper | | Generated by Cas9 RNP electroporation (CRISPR-EZ) |
| Genetic reagent (*Mus musculus*) | B6-$Klra7_{Hss5Δ}$ | This paper | | Generated by Cas9 RNP electroporation (CRISPR-EZ) |
| Antibody | (Armenian hamster monoclonal) anti-CD3e (clone 145–2C11) in PE-Cy5 | Biolegend | RRID: AB_312667 (BioLegend Cat. No. 100302) | FACS (1:400) |
| Antibody | (rat monoclonal) anti-CD4 (clone GK1.5) in BUV737 | BD Biosciences | BD Biosciences Cat# 612844, RRID:AB_2870166 | FACS (1:200) |
| Antibody | (rat monoclonal) anti-CD19 (clone 6D5) in PE-Cy5 | Biolegend | RRID: AB_313644 (BioLegend Cat. No. 115509) | FACS (1:400) |
| Antibody | (rat monoclonal) anti-F4/80 (clone BM8) in PE-Cy5 | Biolegend | RRID: AB_893482 (BioLegend Cat. No. 123112) | FACS (1:400) |
| Antibody | (rat monoclonal) anti-Ter119 (clone TER-119) in PE-Cy5 | Biolegend | RRID: AB_313711 (BioLegend Cat. No. 116210) | FACS (1:400) |
| Antibody | (rat monoclonal) anti-NKp46 (clone 29A1.4) in BV421 | Biolegend | RRID: AB_10915472 (BioLegend Cat. No. 137611) | FACS (1:100) |
| Antibody | (mouse monoclonal) anti-NKG2A$^{B6}$ (clone 16a11) in PE | Biolegend | RRID: AB_10959654 (BioLegend Cat. No. 142803) | FACS (1:50) |
| Antibody | (mouse monoclonal) anti-Ly49A$^{B6}$ (clone A1) in PE | Biolegend | RRID: AB_2134787 (BioLegend Cat. No. 138703) | FACS (1:50) |
| Antibody | (rat monoclonal) anti-NKG2D (clone CX5) in PE-Dazzle 504 | Biolegend | RRID: AB_2728147 (BioLegend Cat. No. 130213) | FACS (1:100) |
| Antibody | (rat monoclonal) anti-CD8β (clone YTS156.7.7) in PE-Cy7 | Biolegend | RRID: AB_2562777 (BioLegend Cat. No. 126616) | FACS (1:200) |
| Antibody | (mouse monoclonal) anti-CD45.1 (clone A20) in APC | Biolegend | RRID: AB_313503 (BioLegend Cat. No. 110714) | FACS (1:200) |
| Antibody | (mouse monoclonal) anti-CD45.2 (clone 104) in FITC | Biolegend | RRID: AB_313443 (BioLegend Cat. No. 109806) Cat. No. 109806 | FACS (1:200) |
| Antibody | (rat monoclonal) anti-CD90.2 (clone 53–2.1) in FITC | Biolegend | RRID: AB_10641145 (BioLegend Cat. No. 140308) | FACS (1:200) |
| Antibody | (rat monoclonal) anti-CD90.2 (clone 53–2.1) in PE | Biolegend | RRID: AB_10641145 (BioLegend Cat. No. 140308) | FACS (1:200) |
| Antibody | (goat polyclonal) anti-mouse IgG (Poly4053) in PE | Biolegend | RRID: AB_315010 (BioLegend Cat. No. 405307) | FACS (1:200) |
| Antibody | (mouse monoclonal) anti-NKG2A (clone 20d5) in PerCP-eFluor 710 | eBioscience/ThermoFisher | RRID: AB_10853352 (Catalog # 46-5896-82) | FACS (1:100) |
| Antibody | (mouse monoclonal) anti-Ly49I (clone YLI-90) in FITC | eBioscience/ThermoFIsher | RRID: AB_2534426 Catalog # A15413 | FACS (1:100) |

*Appendix 1 Continued on next page*

*Appendix 1 Continued*

| Reagent type (species) or resource | Designation | Source or reference | Identifiers | Additional information |
|---|---|---|---|---|
| Antibody | (mouse monoclonal) anti-Ly49G2 (clone 4D11) in PerCP-eFlour 710 | eBioscience/ThermoFIsher | RRID: AB_1834437 Catalog # 46-5781-82 | FACS (1:100) |
| Antibody | (mouse monoclonal) anti-CD90.1 (clone HIS51) in FITC | eBioscience/ThermoFIsher | RRID: AB_465151 Catalog # 11-0900-81 | FACS (1:100) |
| Antibody | (donkey polyclonal) anti-rat IgG F(ab')2 in APC | eBioscience/ThermoFIsher | RRID:AB_469453 polyclonal, lot 17-4822-82 (discontinued) | FACS (1:200) |
| Antibody | (mouse monoclonal) anti-CD8.1 (clone 116–13.1) | BioXCell | RRID: AB_10949065 Catalog # BE0118 | FACS (1:250) |
| Antibody | (rat monoclonal) anti-CD8.2 (clone 2.43) | BioXCell | RRID: AB_1125541 Catalog # BE0061 | FACS (1:50) |
| Antibody | (mouse monoclonal) anti-Ly49A (clone JR9) biotin conjugated | Purified in-house. Ref: Roland and Cazenave *Int. Immunol.* 1992 PMID: 1535510 | | FACS (1:100) |
| Antibody | (mouse monoclonal) anti-Ly49G2[86] (clone 3/25) unconjugated | Used as ascites Ref: Tanamachi et al. *J. Exp. Med.* 2001 PMID: 11157051 | | FACS (1:100) |
| Antibody | (rat monoclonal) anti-NKG2D (clone MI-6) conjugated to biotin in-house | eBioscience/ThermoFIsher | RRID: AB_494129 Catalog # 16-5880-86 | FACS (1:100) |
| Antibody | (rabbit polyclonal) anti-H3K4me1 (ab8895) | Abcam | Abcam Cat# ab8895, RRID:AB_306847 | CUT&RUN (1:50) |
| Antibody | (rabbit polyclonal) anti-H3K4me2 (ab7766) | Abcam | Abcam Cat# ab7766, RRID:AB_2560996 | CUT&RUN (1:50) |
| Antibody | (rabbit polyclonal) anti-H3K4me3 (ab8580) | Abcam | Abcam Cat# ab8580, RRID:AB_306649 | CUT&RUN (1:50) |
| Antibody | (rabbit polyclonal) anti-H3K27ac (ab4729) | Abcam | Abcam Cat# ab4729, RRID:AB_2118291 | CUT&RUN (1:50) |
| Antibody | (rabbit polyclonal) anti-H3K9me3 (ab8898) | Abcam | Abcam Cat# ab8898, RRID:AB_306848 | CUT&RUN (1:50) |
| Antibody | (rabbit monoclonal) anti-H3K27me3 (clone C36B11) | Cell Signaling | Cell Signaling Technology Cat# 4395, RRID:AB_11220433 | CUT&RUN (1:50) |
| Antibody | (rabbit monoclonal) anti-H2AUb1 (clone D27C4) | Cell Signaling | Cell Signaling Technology Cat# 8240, RRID:AB_10891618 | CUT&RUN (1:50) |
| Antibody | (mouse monoclonal) IgG2a k (clone MOPC-173) | Biolegend | Cat # 400202 | CUT&RUN (1:50) |
| Commercial assay or kit | HiScribe T7 Quick High Yield RNA Synthesis Kit | New England Biolabs | Cat # E2050S | |
| Commercial assay or kit | iScript cDNA Synthesis Kit | Bio-rad | Cat # 1708890 | |
| Commercial assay or kit | P3 Primary Cell 4D-Nucleofector X Kit S | Lonza | Cat #: V4XP-3032 | |
| Commercial assay or kit | Mojosort Mouse NK Cell Isolation Kit | Biolegend | Cat # 480049 | |
| Commercial assay or kit | Nextera DNA Library Prep Kit | Nextera | Cat # FC-121–1030 | |
| Commercial assay or kit | NEBNext Ultra II DNA Library Prep Kit for Illumina | New England Biolabs | Cat # E7645S | |
| Commercial assay or kit | NEBNext Multiplex Oligos for Illumina (Index Primers Set 1) | New England Biolabs | Cat # E7335S | |
| Commercial assay or kit | Epitect Fast DNA Bisulfite Kit | Qiagen | Cat # 59,824 | |
| Commercial assay or kit | Invitrogen TOPO TA Cloning Kit for Subcloning, without competent cells | ThermoFisher | Cat # Invitrogen 450641 | |
| Commercial assay or kit | LIVE/DEAD Fixable Near-IR Dead Cell Stain Kit | ThermoFisher | Cat # L34975 | Used at (1:1000) in PBS for flow cytometry |
| Peptide, recombinant protein | Recombinant human IL-2 (teceleukin) | National Cancer Institute (BRB Preclinical Biologics Repository) | | Used at 1,000U/mL for NK cell culture |
| Peptide, recombinant protein | Cas9-NLS (40uM) | UC QB3 MacroLab | | 15.6mg used per nucleofection reaction |
| Peptide, recombinant protein | pA-MNase (Batch #6, 143mg/ml) | Kindly provided by the Henikoff lab | | Used at 0.7mg/mL for CUT&RUN |
| Peptide, recombinant protein | EpiTaq HS (for bisulfite-treated DNA) | Takara | Cat # R110B | |
| Other | DAPI | Biolegend | Cat # 422801 | Used at (1:2000) for flow cytometry |

*Appendix 1 Continued on next page*

*Appendix 1 Continued*

| Reagent type (species) or resource | Designation | Source or reference | Identifiers | Additional information |
|---|---|---|---|---|
| Other | Dynabeads mouse T-activator CD3/CD28 | ThermoFisher | Cat # 11,456D | |
| Other | BioMagPlus Concanavalin A | Bangs Laboratories | Cat # BP531 | |
| Other | AmpureXP beads, 5mL | Beckman Coulter | Cat # A63880 | |
| Software, algorithm | FlowJo Version 10 | FlowJo | https://www.flowjo.com/ RRID:SCR_008520 | |
| Software, algorithm | Python 3.7.4 | Python | https://www.python.org/downloads/release/python-374/ | |
| Software, algorithm | Bowtie 2.1.1 | DOI: 10.1038/nmeth.1923 | | |
| Software, algorithm | Picard | Broad Institute | https://broadinstitute.github.io/picard/ | |
| Software, algorithm | SAMtools 1.8 | DOI:10.1093/bioinformatics/btp352 | | |
| Software, algorithm | BEDTools | DOI: 10.1093/bioinformatics/btq033 | | |
| Software, algorithm | MACS2 | DOI: 10.1186/gb-2008-9-9-r137 | | |
| Software, algorithm | bedGraphToBigWig | DOI: 10.1093/bioinformatics/btq351 | | |
| Software, algorithm | IGV | DOI: 10.1093/bib/bbs017 | | |
| Software, algorithm | SeqPlots | DOI: 0.12688/wellcomeopenres.10004.1 | | |
| Software, algorithm | deepTools | DOI: 10.1093/nar/gku365 | | |

