## [Editor Report]

Kissiov et al., show that enhancers can play an instructive role in controlling stable random monoallelic expression (RME). In order to do so, they initially focus on a limited set of natural killer (NK) receptor genes that are subject to RME, which they investigate using several in vivo genetic models. Furthermore, they show that RME can be considerably more prevalent than previously thought, applying to other gene function than receptors and independently of promoter CpG content. Finally, they provide evidence that enhancer strength and/or number might influence the extent of RME, by affecting the probability of promoter activation. Overall, this is a highly relevant manuscript with major implications in gene regulation and enhancer biology and, thus, of broad scientific interest.

---

## [Decision Letter]

**Decision letter after peer review:**

Thank you for submitting your article "Binary outcomes of enhancer activity underlie stable random monoallelic expression" for consideration by *eLife*. Your article has been reviewed by 3 peer reviewers, and the evaluation has been overseen by a Reviewing Editor and Tadatsugu Taniguchi as the Senior Editor. The following individuals involved in review of your submission have agreed to reveal their identity: Anne-Valerie Gendrel (Reviewer #1); Alvaro Rada-Iglesias (Reviewer #2).

Overall, the reviewers found the manuscript to be relevant, the data clearly presented and the main conclusions well supported by the data. They appreciated the usage of an impressive battery of mouse models, and recognized novel insights brought by this study, into how RME can be regulated and how enhancers can work in a binary manner controlling the probability rather than the level of gene expression. Two slight critics may be (1) the limited mechanistic insights the manuscript provides onto how enhances contribute to RME, and (2) whether the enhancer-RME process described here applies to other types of genes and other cell types, i.e. for genes that are rather CpG-rich than CpG-poor, that are not expressed in the hematopoietic lineage and that do not encode surface markers.

Please refer to the two points further expanded below to prepare your revisions, and please consult the detailed points raised by the reviewers, for further rephrasing of the manuscript and discussion in a point-by-point rebuttal.

1 – Provide additional analyses to investigate the nature of the promoters studied here, before making the claim that RME may be more widespread than originally thought. Using published data, it should be feasible to study these peculiar promoters for their CpG content, and motifs for transcription factor binding sites. In comparison, are genes with CpG-rich promoters also subject to RME and do enhancers influence RME at CpG-rich genes? Finally, although the authors suggest in the discussion that DNA methylation is unlikely to play a major role in RME, it is definitely worth investigating, as an attempt to explain the lack of enhancer-gene communication and also to link with a potential influence on RME extent. Promoters of active and inactive genes should be compared for their DNA methylation states, either using public datasets (if they exist) or using targeted PCR-based DNA methylation analysis.

2 – Moderate the tone regarding the claims that HSSs identified by ATAC-seq are definitely enhancers and not promoters (solely based on epigenomic profiling), provided that this topics is quite controversial in the field.

*Reviewer #1 (Recommendations for the authors):*

Here are my specific comments and questions for the authors to improve the manuscript.

1. Throughout the manuscript, mature NK, immature NK, primary NK, IL-2 expanded NK cells are mentioned in different instances. It is unclear whether this matters or not? Is this important for the expression of the different genes analysed? This should be clarified, especially for non-specialists in these lineages.

What is the stimulation of NK cells with IL-2 important for; is this for proliferation, for activity? Why are cells sometimes expanded with IL-2 and sometimes not?

2. How were the % of expression frequencies on Figure 1—figure supplement 1A calculated?

3. There is no ATAC-seq peak visible for the promoter of the Ly49a gene (Figure 1A, Figure 3A, Figure 4A), unlike for the other genes of the cluster. Could the authors please comment on this? Doesn´t this argue against the claim that HSS1 is an enhancer and not a promoter?

4. On the FACS plot on Figure 2A, it is difficult to distinguish the population of cells expressing the B6 allele only from both alleles. It would help to draw a circle around each cell populations that were selected for the sorting and further analysed for ATAC-seq. Moreover, was there a control of populations after sorting?

5. On Figure 2E, for the control mouse IgG2ak (cIgG) antibody in IL-2 expanded NK cells, there is no ATAC-seq peak at the enhancer (nor CUT&RUN signal). Shouldn't an ATAC-seq peak at the enhancer expected in this population too? Isn't it similar to cells expressing neither alleles?

6. On Figure 3—figure supplement 2C and F. Isn't there a slight increase in the % of cells expressing the BALB only allele in heterozygous F1 mice for both Ly49a and Nkg2a?

7. On lines 344-345, the authors say: "the corresponding region of the highly related Ly49a gene, which is expressed by only ~17% of NK cells, is much less accessible and presumably less active (Figure 4A)". How is the level of accessibility seen/observed? This is not clear to me.

8. Figure 4E – similar to comment #4, isn't the % of cells expressing the BALB allele only increased? …which would be indicative of a slight dosage compensation mechanism?

9. The possible role of the HSS5 enhancer in increasing the expression probability/frequency of the Ly49g gene in NK cells is appealing. Given the similarity between the Ly49a and Ly49g loci regarding the HSS1 enhancer, would it be possible to test this by introducing the HSS5 element upstream of the Ly49a gene to see whether this would increase the probability/frequency of Ly49a expression? (maybe using a transgene construct, not necessarily a knockin).

10. Line 442; there is no introduction for the knockout model of Nkg2d. What is the KO allele exactly? What is deleted: promoter, exons, whole locus?

11. Line 464; The authors mention the 3'enhancer for the Nkg2d gene. Could they speculate on its function?

12. Line 573-575; the following sentence is not clear to me: "Importantly, NKG2D- cells in both Nkg2d5'ED/5'ED and Nkg2d+/- mice were as likely to express NKG2A, Ly49G2 or Ly49I as NKG2D+ cells, suggesting stochastic expression of the WT Nkg2d allele (Figure 7—figure supplement 1, A-C)." Could the authors please clarify?

13. The following two important points remain unanswered in this study and could be discussed further in the discussion part of the manuscript:

a – What define the strength of an enhancer for the Ly49 and Nkg2 genes. What is the difference between a weak and a strong enhancer? Related to the binding of specific transcription/chromatin factors? Level of enrichment of specific histone marks? 3D conformation?

b – Could this model apply to chromatin or nuclear factors, expressed in other cell lineages than the hematopoietic lineage?

*Reviewer #2 (Recommendations for the authors):*

Overall, I found the manuscript highly relevant and the main conclusions are well supported by the presented data. However, there are some aspects that could be modified and/or expanded in order to improve the overall relevance of the work:

My main concern is that, as presented, the work strongly suggest that enhancers play an important and instructive work in RME. However, there are limited insights into the molecular mechanisms behind such an instructive role. Given that the promoter but not the enhancer chromatin state correlates with RME, I think the authors should explore more how enhancer-gene communication might be altered in expressing and non-expressing alleles. Since the investigated enhancers are quite proximal to the target gene promoters, investigating topological aspects of enhancer-gene communication can be technically quite challenging. Moreover, it is highly intriguing that in alleles subject to RME the promoters stably lose their responsiveness to proximal active enhancers. In this regard, I have a few suggestions:

– Although H3K27ac is strongly associated with active enhancers, it can be uncoupled from expression of the target genes (Pachano et al., 2021), while the production of eRNAs seems tope strongly coupled with mRNA expression (Kim et al., 2010; Hirabayashi et al., 2019). Therefore, I think it could be quite relevant to investigate eRNA levels at the investigated enhancers in the context of expressing and non-expressing alleles.

– As the authors clearly show, the promoters of the NK receptor genes do not acquire histone modifications associated with gene repression. This is not surprising given that, as the authors mention in the discussion, these promoters are CpG-poor. Inactive CpG-poor promoters do not typically show enrichment for these histone marks but display high DNA methylation levels instead, in contrast with CpG-rich promoters, which are typically hypomethylated regardless of whether they are active or not. Therefore, although the authors mention in the Discussion that DNA methylation is unlikely to play a major role in RME, I still think that it would be quite insightful to compare the DNA methylation levels of active and inactive promoters for example by traditional locus-specific bisulfite sequencing. Moreover, the authors could investigate whether their findings still apply to genes with CpG-rich promoters (are CpG-rich genes subject to RME? Do enhancers influence RME at CpG-rich genes?) and/ or, alternatively, introduce a CpG island into one of the NK receptor gene promoters.

– If the authors find clear differences in the DNA methylation status of active and inactive promoters, one possibility is that epigenetic heterogeneity at gene promoters might occur before enhancer activation and thus influence, together with enhancer strength, the extent of RME. The authors could investigate DNA methylation at promoter regions at earlier cellular states, which could represent a critical temporal window at which DNA demethylating agents could have an impact on RME.

*Reviewer #3 (Recommendations for the authors):*

1 – In the first section of the results, the authors test the idea that Ly49Hss and NKg25'E regulatory elements are enhancers rather than promoters in NK cells as previously suggested. A full demonstration of this is not provided since the data presented is essentially epigenetic features and ATAC-seq. The provided evidence does not for example demonstrate that Ly49Hss can not behave as promoters in immature cells as proposed before since no RNA data is shown in the article for NKR analyses.

2 – In the same section, based on epigenetic profiles and the relative amount of H3K4me1/3 and other predictive methods (ATAC, K27ac) the authors formally conclude on 'enhancer identity' (lines 186-187) and 'these findings provide definitive support for the conclusion that Ly49Hss1 and Nkg25'E elements represent enhancers' (lines 196-197). It is important to keep in mind that epigenomic features never demonstrate enhancer identity but rather suggest of putative enhancer features. Thus, the phrasing used here should be toned down.

3 – For all ATAC-seq and ChIP-seq data shown, including examples the units should appear on the graphs as normalized equivalently to the read counts (for example in Figure 1B, Figure 1s2C and all the following). A table indicating all published and original data sets used with GSE numbers should be provided, including replicates used and sequencing depth. The presented novel data does not seem to have been submitted to GEO database. This should be done.

4 – If the model proposed by the authors is correct, it implies a level of stochasticity in promoter-enhancer dependent activation. Based on sequence elements (for example presence or absence of CpG islands), do the promoters of the NKR locus tend to be less constitutive? More generally how do the authors envision the critical parameters limiting promoter's action and their strong dependency to enhancer stochastic activation? These points could be discussed in the Discussion section.

---

## [Author Response]

1 – Provide additional analyses to investigate the nature of the promoters studied here, before making the claim that RME may be more widespread than originally thought. Using published data, it should be feasible to study these peculiar promoters for their CpG content, and motifs for transcription factor binding sites. In comparison, are genes with CpG-rich promoters also subject to RME and do enhancers influence RME at CpG-rich genes? Finally, although the authors suggest in the discussion that DNA methylation is unlikely to play a major role in RME, it is definitely worth investigating, as an attempt to explain the lack of enhancer-gene communication and also to link with a potential influence on RME extent. Promoters of active and inactive genes should be compared for their DNA methylation states, either using public datasets (if they exist) or using targeted PCR-based DNA methylation analysis.

We address these concerns in detail in the response to the reviewers’ comments below. In summary, RME can be found at genes with both high and low CpG content (Eckersley-Maslin et al., *Dev. Cell* 2014 PMID: 24576421). As one example from our work, in Figure 7, we show random monoallelic expression of the *Thy1* gene, which is expressed by T cells and is widely regarded as a T cell “marker”. As documented in the UCSC genome browser screenshot depicted in reviewer data, Figure 2 (embedded in the response to reviewer 2) the *Thy1* gene includes a CpG island at its promoter. Furthermore, the *Bcl11b* gene, which encodes a transcription factor that commits thymocytes to the T cell fate, contains a promoter CpG-island and was recently shown to be an RME gene and whose expression probability is further reduced by enhancer-deletion (Ng et al., *eLife* 2018 PMID: 30457103). Therefore, we do not believe there is reason to believe that our results are unique to a class of genes with respect to CpG density; rather, the RME phenomenon (as well as the enhancer deletion-associated variegation phenomenon) appears to apply across cell types and gene classes.

Concerning DNA methylation, we have extensively revised the manuscript to address this issue and will discuss this below in response to reviewer 2.

2 – Moderate the tone regarding the claims that HSSs identified by ATAC-seq are definitely enhancers and not promoters (solely based on epigenomic profiling), provided that this topics is quite controversial in the field.

We softened the language, but note that our data provide evidence beyond epigenomic profiling that these elements are enhancers in mature NK cells: specifically, we showed that deletion of Hss1 in mature NK cells extinguishes expression, which is known to initiate at promoters well downstream of the enhancers in mature NK cells.

Reviewer #1 (Recommendations for the authors):Here are my specific comments and questions for the authors to improve the manuscript.1. Throughout the manuscript, mature NK, immature NK, primary NK, IL-2 expanded NK cells are mentioned in different instances. It is unclear whether this matters or not? Is this important for the expression of the different genes analysed? This should be clarified, especially for non-specialists in these lineages.

Where relevant, we have attempted to explain the characteristics of each type of NK cell. Briefly, immature NK cells are not yet terminally differentiated and importantly do not yet express Ly49 genes, expression of which is associated with mature, or fully differentiated NK cells. This is now indicated on page 10.

What is the stimulation of NK cells with IL-2 important for; is this for proliferation, for activity? Why are cells sometimes expanded with IL-2 and sometimes not?

Generally, we use IL-2 to expand NK cells when it is necessary obtain sufficient cells for analysis, which we now indicate in the text on page 15. To elaborate, NK cells are fairly rare among spleen cells (3%) and only 3 % of those co-express both alleles of Ly49G2 in (B6 x BALB/c)F_1_ hybrid mice, for a frequency of 0.09% of splenocytes. Generally, CUT&RUN experiments utilized IL-2 expanded cells, while ATAC-seq experiments utilized fresh cells. These data appear to cross-validate each other well (Figure 2). Furthermore, Ly49 expression states (both on and off) are stable in NK cells expanded in IL-2. For these reasons, we used IL-2 expanded NK cells when we needed larger numbers of primary cells for chromatin assays.

2. How were the % of expression frequencies on Figure 1—figure supplement 1A calculated?

The percentages are based on previously published historical data generated by both our group and others. We have added citations in the Figure 1—figure supplement 1 legend in the revised manuscript. The numbers depict approximations only.

3. There is no ATAC-seq peak visible for the promoter of the Ly49a gene (Figure 1A, Figure 3A, Figure 4A), unlike for the other genes of the cluster. Could the authors please comment on this? Doesn´t this argue against the claim that HSS1 is an enhancer and not a promoter?

The transcription start sites (TSSs) of the *Ly49* genes were previously documented by 5’ oligo-capping RACE (Gays et al., *PLos One* 2011 PMID: 21483805), and for the Ly49 genes these range from ~5-6kb *downstream* of HSS1, at sites called “Pro2” and “Pro3,” hereafter referred to as the TSSs. The citation is now included in the Figure 2 legend. mRNAs in mature, Ly49+ NK cells do NOT initiate at the HSS1 elements, and this is true across the gene family. Thus, HSS1 is not the promoter for *Ly49a* or any of the *Ly49* genes in mature NK cells, nor has any published report made the claim that it is. While the signals from the documented TSSs for *Ly49a* are weak, they are present (see Author response image 1). The weakness of these ATAC sites is likely due to (1) Multiple TSSs such that the ATAC-seq accessibility is diffused to several sites, and (2) the fact that less than 20% of cells express the gene. The signal would therefore be ~5 times higher (or more) if the cells were sorted to be Ly49A+.

**Author response image 1. sa2fig1:** ATAC-seq tracks in NK cells at the *Ly49a* locus show weak signal at the *Ly49a_Hss5_* and *Ly49a_Pro_* elements. Here we display four ATAC-seq tracks generated from (B6 x BALB/c)F_1_ NK cells sorted according to Ly49G2 allelic expression status as shown in Figure 2D, but at the *Ly49a* locus. The locations of *Hss1*, *Hss5*, and the major annotated TSSs are indicated.

4. On the FACS plot on Figure 2A, it is difficult to distinguish the population of cells expressing the B6 allele only from both alleles. It would help to draw a circle around each cell populations that were selected for the sorting and further analysed for ATAC-seq. Moreover, was there a control of populations after sorting?

The NKG2A^B6^-specific mAb (16a11) was first generated in our lab (Vance et al. *PNAS* 2002 PMID 11782535), and it was subsequently shown by Dixie Mager’s lab that the use of this clone in combination with the NKG2A^B6+BALB/c^ reactive clone 20d5 enables the sorting the populations expressing each separately, which was confirmed at the mRNA level (Rogers et al., *J. Immunol.* 2006 PMID: 16785537) and is now cited on p. 11. We added a new figure (Figure 2 figure supplement 1) that shows the gating and post sort analysis of the NKG2A-sorted cells and the Ly49G2-sorted cells used to generate ATAC-seq data and the CUT&RUN data presented in Figure 2.

5. On Figure 2E, for the control mouse IgG2ak (cIgG) antibody in IL-2 expanded NK cells, there is no ATAC-seq peak at the enhancer (nor CUT&RUN signal). Shouldn't an ATAC-seq peak at the enhancer expected in this population too? Isn't it similar to cells expressing neither alleles?

The control IgG “cIgG” refers to the antibody used as a control for the CUT&RUN itself, not as a control for sorting; we do not expect to observe a signal with an IgG that should not interact with the chromatin. We regret the omission of a clear explanation. The cells used for the cIgG control are a mixture of NK cells that express either the B6 or BALB/c allele, in approximately equal amounts. In order to clarify this, we changed the coloring of the cIgG label to black to distinguish it from the labeling of the allele expressed. We further explain in the revised legend for Figure 2 the nature of this control experiment and specify which cells were used to generate the cIgG track. The protein A in the pA-Mnase fusion binds mIgG2ak with high affinity, which is the reason for the selection of this control. In general, CnR produces remarkably low noise, and in all cIgG CnR datasets we have produced we have not seen appreciable signal enrichment at ATAC-seq accessible sites.

6. On Figure 3—figure supplement 2C and F. Isn't there a slight increase in the % of cells expressing the BALB only allele in heterozygous F1 mice for both Ly49a and Nkg2a?

In each of the referenced panels, we *expect* to see an increase in frequency of BALB/c only cells without invoking a compensation mechanism. This is not due to an increased likelihood of expression of the BALB/c allele, but rather because the cells that would have expressed both alleles are no longer able to express the B6 allele as they lack the critical enhancer on the B6 chromosome. In other words, while the proportion of cells expressing *only* the BALB/c allele increases, the total proportion of cells expressing the BALB/c allele remains unchanged.

If the reviewer is instead referring to the second sub-panel in Figure 3—figure supplement C (where there is a slight increase in the observed BALB/c-only cells over expected) this is a separate point. We *expect* an increase in the BALB/c only cells for reasons entirely independent of a compensatory effect, as described above. However, this slight increase in observed over expected may imply some slight compensatory mechanism. This increase is slight, and the overall result is that the total number of cells expressing the BALB/c allele is not drastically increased as we might expect if there were a trans-acting transcriptional “counting” mechanism to ensure a particular percentage of cells express NKG2A.

7. On lines 344-345, the authors say: "the corresponding region of the highly related Ly49a gene, which is expressed by only ~17% of NK cells, is much less accessible and presumably less active (Figure 4A)". How is the level of accessibility seen/observed? This is not clear to me.

Figure 4A shows that the region of *Ly49a* that aligns with Hss5 of Ly49g has only a very weak ATAC signal. This was true for all ATAC-seq datasets generated in NK cells that we have examined. The ATAC-seq data in Reviewer Data Figure 1 also shows a weak Hss5 signal that is barely distinguishable above background, and certainly much less detectable than *Ly49g-*Hss5 (Figure 2D; Figure 4A).

8. Figure 4E – similar to comment #4, isn't the % of cells expressing the BALB allele only increased? …which would be indicative of a slight dosage compensation mechanism?

See comment 6. Cells that would otherwise express both alleles in a WT mouse that fail to express the B6 allele in the mutant F1 are now identified as BALB only cells, resulting in an increased proportion without any compensation mechanism.

If, again, the reviewer is referring to the slight increase in observed BALB/c-only cells as compared to expected, the effect is even more slight here and is much more consistent with stochastic and independent regulation of alleles rather than a compensatory mechanism that acts in trans.

9. The possible role of the HSS5 enhancer in increasing the expression probability/frequency of the Ly49g gene in NK cells is appealing. Given the similarity between the Ly49a and Ly49g loci regarding the HSS1 enhancer, would it be possible to test this by introducing the HSS5 element upstream of the Ly49a gene to see whether this would increase the probability/frequency of Ly49a expression? (maybe using a transgene construct, not necessarily a knockin).

This is potentially a revealing experiment that we would love to do and plan to do. In order to do it properly, it should be a germline knockin. Transgenes suffer from various technical problems including variable copy number insertions and position effects. We previously successfully used a genomic transgene to establish the proximal regulation of *Ly49a* (Tanamachi et al., *J. Immunol.* 2004 PMID: 14707081). The data, however, were variable for different founders, to an extent that we believe would compromise this approach for the question at hand. Knockin approaches for these genes have proven to be extremely challenging in our hands likely due to the homology between the targeting sequence and multiple related *Ly49* genes which appear to diminish the frequency of replacement of the desired gene. For these reasons we decided this experiment, while potentially revealing, will be subject of future efforts and, if successful, included in a future publication.

10. Line 442; there is no introduction for the knockout model of Nkg2d. What is the KO allele exactly? What is deleted: promoter, exons, whole locus?

The *Nkg2d* knockout model was generated and characterized by our group (Guerra et al., *Immunity* 2008 PMID: 18394936). The gene is comprised by 7 exons, of which exons 2-6 are deleted in the knockout model. We added a statement to this effect and a reference on p. 27 of the revised manuscript.

11. Line 464; The authors mention the 3'enhancer for the Nkg2d gene. Could they speculate on its function?

We added a sentence on p 28 to note the possibility that the 3’ element constitutes the residual enhancer that drives expression after 5’E deletion, and which acts in combination with 5’E to drive a very high probability of *Nkg2d* expression in NK cells. However, we have no direct evidence at present that the 3’ ATAC-seq peak is actually an enhancer of *Nkg2d*.

12. Line 573-575; the following sentence is not clear to me: "Importantly, NKG2D- cells in both Nkg2d5'ED/5'ED and Nkg2d+/- mice were as likely to express NKG2A, Ly49G2 or Ly49I as NKG2D+ cells, suggesting stochastic expression of the WT Nkg2d allele (Figure 7—figure supplement 1, A-C)." Could the authors please clarify?

We revised this sentence (p. 33) to lend more clarity. It now states:

“Importantly, in both *Nkg2d^5′E^*^D/*5′E*D^ and *Nkg2d^+^*^/*-*^ mice, NKG2D-negative cells were as likely as NKG2D+ cells to express NKG2A, Ly49G2 or Ly49I (Figure 7—figure supplement 1, A-C). These data suggest that in both knockout and WT mice, NKG2D is randomly expressed with respect to the other RME receptor genes.”

13. The following two important points remain unanswered in this study and could be discussed further in the discussion part of the manuscript:a – What define the strength of an enhancer for the Ly49 and Nkg2 genes. What is the difference between a weak and a strong enhancer? Related to the binding of specific transcription/chromatin factors? Level of enrichment of specific histone marks? 3D conformation?

We provide a speculative answer in our response to the public reviews and in the revised manuscript on p. 43. On p 43 of the revised manuscript, after proposing that the randomness of RME may reflect the probability of overcoming the energy barrier required for nucleosome remodeling of initially chromatinized promoters, we added the following passage:

“It may be this property that is the measure of “enhancer strength”, i.e., the capacity of the enhancer to overcome the energy threshold presented by the promoter within a limited window during differentiation. The number of enhancer elements in a gene, the availability of relevant transcription factors that bind the enhancers and the affinity with which the enhancer binds the factors are all likely relevant in determining enhancer strength.”

b – Could this model apply to chromatin or nuclear factors, expressed in other cell lineages than the hematopoietic lineage?

We believe the answer is yes. As already mentioned, the gene encoding the transcription factor Bcl11b is expressed in an RME fashion during thymocyte development, and the probability of expression of this gene is further altered by enhancer deletion variegation (Ng et al., *eLife* 2018 PMID: 30457103). We add a reference to this in this discussion on page 41, indicating that this supports the generality of our model to genes not encoding cell surface receptors. With respect to non-hematopoietic lineages, deletion of enhancers regulating the odorant receptor genes (Khan et al., *Cell* 2011 PMID: 22078886) as well as trace amine-associated and trace-amine associated receptors (Fei et al., *Nat. Commun.* 2021 PMID: 34145235) in the olfactory epithelium reduced the expression probability of the target genes, demonstrating that RME rooted in binary enhancer action applies to non-hematopoietic lineage. On page 44 in the revised manuscript, we add statements and citations to these papers in order to clarify that the mechanisms we describe are likely not restricted to hematopoietic cells.

Reviewer #2 (Recommendations for the authors):Overall, I found the manuscript highly relevant and the main conclusions are well supported by the presented data. However, there are some aspects that could be modified and/or expanded in order to improve the overall relevance of the work:My main concern is that, as presented, the work strongly suggest that enhancers play an important and instructive work in RME. However, there are limited insights into the molecular mechanisms behind such an instructive role. Given that the promoter but not the enhancer chromatin state correlates with RME, I think the authors should explore more how enhancer-gene communication might be altered in expressing and non-expressing alleles. Since the investigated enhancers are quite proximal to the target gene promoters, investigating topological aspects of enhancer-gene communication can be technically quite challenging. Moreover, it is highly intriguing that in alleles subject to RME the promoters stably lose their responsiveness to proximal active enhancers. In this regard, I have a few suggestions:– Although H3K27ac is strongly associated with active enhancers, it can be uncoupled from expression of the target genes (Pachano et al., 2021), while the production of eRNAs seems tope strongly coupled with mRNA expression (Kim et al., 2010; Hirabayashi et al., 2019). Therefore, I think it could be quite relevant to investigate eRNA levels at the investigated enhancers in the context of expressing and non-expressing alleles.

We appreciate the close relationship between eRNA and target mRNA production. While we have not directly assayed eRNAs ourselves, they have been analyzed extensively by RT-PCR in both primary NK cells and NK cell related cell lines (Gays et al., *J. Immunol.* 2015 PMID: 25926675) (cited on p. 9 of revised manuscript). This study showed that bidirectional enhancer-derived transcripts are present in cells that do not express the corresponding receptor gene. Therefore, even when enhancer-derived transcripts are used to measure enhancer activity, Hss1 activity can be decoupled from target gene expression status, corroborating our results. Taken together, these results suggest that the relationship between the enhancer activity acting on a locus is fundamentally probabilistic rather than deterministic (i.e., not 1:1). This decoupling likely takes place infrequently, when overall enhancer activity is below the threshold to activate the target gene in the large majority of cells. In many and perhaps most cases, enhancer activity as measured by eRNA production will correlate very strongly with mRNA production, especially when measured in bulk populations rather than single cells. We added a sentence on p. 14 addressing the relationship between eRNA production and target *Ly49* gene expression status.

– As the authors clearly show, the promoters of the NK receptor genes do not acquire histone modifications associated with gene repression. This is not surprising given that, as the authors mention in the discussion, these promoters are CpG-poor. Inactive CpG-poor promoters do not typically show enrichment for these histone marks but display high DNA methylation levels instead, in contrast with CpG-rich promoters, which are typically hypomethylated regardless of whether they are active or not. Therefore, although the authors mention in the Discussion that DNA methylation is unlikely to play a major role in RME, I still think that it would be quite insightful to compare the DNA methylation levels of active and inactive promoters for example by traditional locus-specific bisulfite sequencing. Moreover, the authors could investigate whether their findings still apply to genes with CpG-rich promoters (are CpG-rich genes subject to RME? Do enhancers influence RME at CpG-rich genes?) and/ or, alternatively, introduce a CpG island into one of the NK receptor gene promoters.

We regret that we did not dedicate more space in the first version of the paper to the discussion of DNA methylation. We have added data and much discussion on the topic, including a section in the results beginning on p 28 and supplementary figure (Figure 5—figure supplement 2) showing differential methylation of the *Nkg2d* promoter on active and silent alleles in *Nkg2d5’E* deletion mice. We further revised the discussion to address the role of DNA methylation in RME beginning on p44. Briefly, many RME genes do not display differential promoter methylation (da Rocha, Gendrel *Essays Biochem.* PMID: 31782494). Recent studies have further indicated that only a small minority of RME genes are subject to derepression of silent alleles through pharmacological inhibition of DNA methylation (Gupta et al., *G3* 2021 https://doi.org/10.1093/g3journal/jkab428 ; Marion-Poll et al., *Nat. Commun.* 2021 PMID: 34504093). Furthermore, inhibition of DNA methylation did not derepress silent *Igh* alleles in a B cell hybridoma model of enhancer-deletion variegation (Ronai et al., *J. Immunol.* 2002 PMID: 12471125). The role of DNA methylation in RME was addressed in a recent review (da Rocha, Gendrel *Essays Biochem.* PMID: 31782494), and it appears to be accepted in the RME field that DNA methylation is unlikely to underly all or most cases of RME (Eckersley-Maslin, Spector *Trends Genet.* 2014 PMID: 24780084). As we mention in the discussion (page 43), there remains the possibility that DNA methylation may play a role in determining the probability of promoter activation for some RME genes, including the ones we studied in the paper.

Regarding the issue of CpG density: *Ly49* promoters are indeed CpG-poor, but it is the case that CpG-rich genes are also subject to RME (Eckersley-Maslin et al., *Dev. Cell* 2014 PMID: 24576421). Indeed, the *Thy1* gene, which we show is an RME gene despite being considered a marker of all T cells, harbors a CpG island annotated on the UCSC genome browser (Author response image 2). The RME *Bcl11b* (Ng et al., *eLife* 2018 PMID: 30457103) also harbors a CpG island (Author response image 3). We add a statement in the discussion beginning (p. 42) to clarify that RME (and enhancer-controlled RME) apply to genes with CpG-dense and CpG poor promoter regions.

**Author response image 2. sa2fig2:** The *Thy1* gene harbors a promoter proximal CpG island. We provide a screenshot of the UCSC genome browser displaying the *Thy1* locus with the CpG track. The red box highlights the region of interest. We show in Figure 7 of the manuscript that Thy1, which is widely regarded as a marker of all T cells, is actually an RME gene. CpG islands are defined as those with GC content of >50%, and the ratio of observed/expected CpG dinucleotides is >0.6, as described in the following link: https://genome.ucsc.edu/cgi-bin/hgc?db=mm10&c=chr9&l=44043001&r=44048197&o=44043488&t=44043701&g=cpgIslandExt&i=CpG%3A+16.

**Author response image 3. sa2fig3:** The *Thy1* gene harbors a promoter proximal CpG island. Similar to Author response image 2, we provide a UCSC genome browser screenshot of the *Bcl11b* locus with the CpG track displayed. It was recently shown that the gene, which encodes a key transcription factor commiting thymocytes to the T cell lineage, is expressed in an RME fashion and is further subect to enhancer deletion-associated variegation (Ng et al., *eLife* 2018 PMID: 30457103). The promoter region of interest is highlight in the red box, and the CpG island is indicated with the red arrow.

– If the authors find clear differences in the DNA methylation status of active and inactive promoters, one possibility is that epigenetic heterogeneity at gene promoters might occur before enhancer activation and thus influence, together with enhancer strength, the extent of RME. The authors could investigate DNA methylation at promoter regions at earlier cellular states, which could represent a critical temporal window at which DNA demethylating agents could have an impact on RME.

With respect to the methylation of CpG sites in the *Ly49* and *Nkg2a* promoters themselves, several studies have been published by Dixie Mager’s group demonstrating a clear correlation between DNA methylation and silent alleles (Rouhi et al., *J. Immunol.* 2006 PMID: 16493057; Rouhi et al., *Mol. Immunol.* 2007 PMID: 16750269; Rogers et al., *J. Immunol*. 2006 PMID: 16785537). Inhibition of DNA methylation on its own appears insufficient to derepress silent alleles, however. In the case of *Nkg2a*, the promoter region is heavily methylated both in immature cells that do not yet express NKG2A, and in the case of silent alleles in mature cells, suggesting that methylation is not specifically deposited on silent alleles, but rather that active alleles are de-methylated. This is mentioned in the revised discussion (page 43). Furthermore, the *Ly49a* promoter region (Pro2) was shown to be methylated in non-hematopoietic cells. All of these data support the notion that DNA methylation of the NK receptor gene promoters is associated with the default off state, and that demethylation takes place specifically on activated alleles, likely concomitant with activation. These considerations make it less likely that there exists an immature developmental state in which the two alleles might be differentially methylated, preparatory of subsequent RME. Another important point is that the enhancer deletion variegation (and associated allelic “failure”) was observed in *Drosophila* (Perry et al., *PNAS* 2011 PMID: 21825127), where DNA methylation is not thought to be present at detectable levels and thus does not play a major role in gene regulation. This consideration confirms that DNA methylation is not generally required for the binary enhancer-driven RME phenomenon.

As discussed in the revised Discussion section (p 42), we favor a mechanism in which enhancer action must overcome the energy barrier associated with the promoters being initially in an inactive state, and success occurs with a random probability, albeit dependent on enhancer strength. In our thinking, the inactive state that must be overcome could simply be the closed chromatin state. However, we also propose the possibility that promoter DNA methylation could contribute to the energy barrier in some cases.

Reviewer #3 (Recommendations for the authors):1 – In the first section of the results, the authors test the idea that Ly49Hss and NKg25'E regulatory elements are enhancers rather than promoters in NK cells as previously suggested. A full demonstration of this is not provided since the data presented is essentially epigenetic features and ATAC-seq. The provided evidence does not for example demonstrate that Ly49Hss can not behave as promoters in immature cells as proposed before since no RNA data is shown in the article for NKR analyses.

As requested, we toned down the statements claiming the chromatin modification data provided “definitive” support for enhancer as opposed to promoter identity (p. 10). Nevertheless, we feel that the evidence is strong. Our argument depends not only on the epigenetic marks and ATACseq results, but also on our finding that the elements are necessary for maintaining gene expression in mature NK cells, where it is known that the mRNAs are initiated well downstream of Hss1 (Figure 1—figure supplement 2). The revised text on p. 10-11 emphasizes this point (see our response to reviewer 1, comment 3 for further details). The reviewer points out that we did not examine RNA, but others have, and we do not contest their findings that transcripts initiating at Hss1 and transcribing through the gene have been identified in immature NK cells. The difficulty is that transcripts (eRNAs) are known to initiate from enhancers, so the RNA data does not resolve the issue. Another group has made a good argument that the observed transcripts initiating at Hss1 are indeed eRNAs (Gays et al., *J. Immunol.* 2015 PMID: 25926675). We note that the existence of eRNAs in other genes has not led to theories that such RNAs are part of a developmental switch that turns those genes on and off. An additional argument in favor of our conclusions is that deletion of the 5’ enhancer of *Nkg2d* resulted in an expression pattern indistinguishable from that of *Ly49* or *Nkg2a* genes. Since it is unlikely that enhancer deletion would CREATE a bidirectional promoter that drives variegated expression, the simplest hypothesis is that variegation in Ly49 and enhancer deletion variegation alike is the consequence of weak enhancer activity as opposed to the action of a bidirectional upstream promoter.

2 – In the same section, based on epigenetic profiles and the relative amount of H3K4me1/3 and other predictive methods (ATAC, K27ac) the authors formally conclude on 'enhancer identity' (lines 186-187) and 'these findings provide definitive support for the conclusion that Ly49Hss1 and Nkg25'E elements represent enhancers' (lines 196-197). It is important to keep in mind that epigenomic features never demonstrate enhancer identity but rather suggest of putative enhancer features. Thus, the phrasing used here should be toned down.

See our response to the previous comment, where we describe the modifications we made and their basis.

3 – For all ATAC-seq and ChIP-seq data shown, including examples the units should appear on the graphs as normalized equivalently to the read counts (for example in Figure 1B, Figure 1s2C and all the following). A table indicating all published and original data sets used with GSE numbers should be provided, including replicates used and sequencing depth. The presented novel data does not seem to have been submitted to GEO database. This should be done.

We have added the units on the graphs as requested. We have submitted to GEO and provided the accession number: GSE181197. The secure token avcpeismhpehlgd is required to access the series prior to publication. Furthermore, we have provided the metadata for the GEO series, with an added column indicating sequencing depth for each file as the file “220302_GEO submission metadata.”

4 – If the model proposed by the authors is correct, it implies a level of stochasticity in promoter-enhancer dependent activation. Based on sequence elements (for example presence or absence of CpG islands), do the promoters of the NKR locus tend to be less constitutive? More generally how do the authors envision the critical parameters limiting promoter's action and their strong dependency to enhancer stochastic activation? These points could be discussed in the Discussion section.

As noted in our response to Reviewer 2’s comments, we have now expanded the Discussion section to include discussion of our conception of the critical parameters that determine stochasticity. Some of these (enhancer strength) come from our work and others, such as the abundance of relevant transcription factors, come from the work of others, which we cite.

In conclusion, we have made substantial changes to the manuscript to address the comments of all the reviewers. We believe the manuscript is improved as a result.